# EMPIRICAL STUDY ON ENHANCING EFFICIENCY IN MASKED IMAGE MODELING PRE-TRAINING

## ABSTRACT

The combination of transformers and masked image modeling (MIM) pre-training framework has shown remarkable potential in various vision tasks. However, the high computational cost of pre-training hinders the practical application of MIM. This paper introduces *FastMIM*, a simple and versatile framework that expedites masked image modeling through two steps: (i) pre-training vision backbones using low-resolution input images and (ii) reconstructing Histograms of Oriented Gradients (HOG) feature instead of original RGB values of the input images. Furthermore, we propose *FastMIM-P*, which progressively increases the input resolution during the pre-training stage to improve the transfer learning performance of models with high capacity. We point out that: (i) a wide range of input resolutions during pre-training can result in similar performances in fine-tuning and downstream tasks such as detection and segmentation; (ii) the shallow layers of encoder are more important during pre-training, and discarding the last few layers can speed up the training process without affecting fine-tuning performance; and (iii) HOG is more stable than RGB values when transferring resolution. Equipped with *FastMIM*, any type of vision backbone can be efficiently pre-trained. For example, using ViT-B/Swin-B as backbones, we achieve 83.8%/84.1% top-1 accuracy on ImageNet-1K. Compared to previous approaches, our method can achieve better top-1 accuracy while accelerating the training procedure by ∼5×.

## 1 INTRODUCTION

Self-supervised learning is a promising paradigm that aims to learn feature representations from scalable unlabeled data, and has achieved significant results in natural language processing (NLP) through masked language modeling (MLM) (Radford et al., 2018; Devlin et al., 2018; Brown et al., 2020; Chen et al., 2020b). Recently, it has also attracted increasing attention in vision community, where masked image modeling (MIM) has emerged as a self-supervised pre-training framework. Different from previous contrastive learning based approaches (Wu et al., 2018; Chen et al., 2020c; He et al., 2020; Caron et al., 2021), MIM learns representations through a mask-then-predict manner, *e.g.*, predicting the raw pixels (He et al., 2021; Xie et al., 2022) or other tokenizations (Bao et al., 2021; Zhou et al., 2021; Chen et al., 2022b) of randomly masked input images.

Despite recent achievements and the state-of-the-art results on various downstream vision tasks, the pre-training stage of self-supervised learning-based approaches is extremely ***computationally expensive*** and ***slow***. For example, contrastive learning based SimCLR (Chen et al., 2020c) takes fifteen hours on 128 TPU v3 cores (1920 TPU hours in total) to finish the 1000 epochs training on ResNet-50 (He et al., 2016) with a batch size of 4096. Moreover, MIM based BEiT (Bao et al., 2021) takes about five days using 16 32GB V100 GPUs (1920 GPU hours in total, not counting the time for

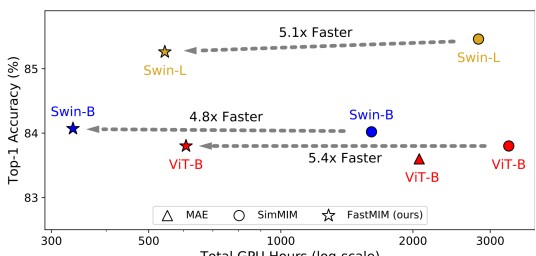

Figure 1: Comparisons in terms of total GPU hours (pre-training time) on ImageNet-1K classification task. *FastMIM* expedites the pre-training stage by ∼5×.

dVAE (Rolfe, 2016; Van Den Oord et al., 2017) pre-training) to accomplish 800 epochs training on ViT-B (Dosovitskiy et al., 2020). To pre-train vision backbones efficiently, He *et al.* proposes

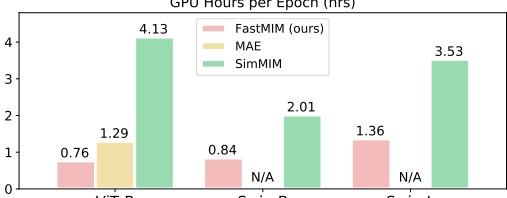 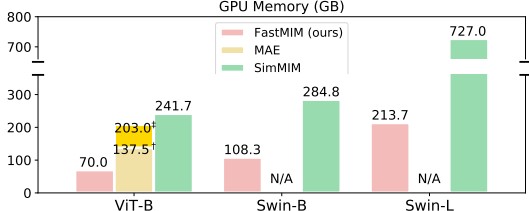

Figure 2: Comparison of our *FastMIM*, MAE (He et al., 2021) and SimMIM (Xie et al., 2022) in terms of GPU efficiency. All frameworks use a ViT-B/Swin-B/Swin-L encoder and a batch size of 2048. The experiments are conducted on a single machine with 8 32GB V100 GPUs. [†]: MAE decoder has 1 block (1b512d). [‡]: MAE decoder has 8 blocks (8b512d). N/A: MAE is not suitable for Swin (Liu et al., 2021).

the masked autoencoder (MAE) (He et al., 2021) which discards the masked tokens and only operates on the whole input sequences in the lightweight decoder. Notably, although this asymmetric encoder-decoder design significantly reduces the computational burden, MAE can only support the isotropic transformer architecture (Dosovitskiy et al., 2020), withholding it from becoming a ***generic*** MIM framework for various vision backbones (Wang et al., 2021; Liu et al., 2021; Guo et al., 2022; Chu et al., 2021). In contrast to above discarding strategy, SimMIM (Xie et al., 2022) retains both visible and masked tokens. In this way, SimMIM can be naturally applied to different models, *e.g.*, isotropic ViT (Dosovitskiy et al., 2020) and hierarchical Swin Transformer (Liu et al., 2021). However, it suffers from heavy memory consumption that even the base size model such as Swin-B cannot be trained via SimMIM framework on a single machine with 8 32GB V100 GPUs (Huang et al., 2022).

To reduce the pre-training costs of self-supervised learning and make MIM a more ***efficient*** and ***practicable*** framework for vision, we devise a simple and straightforward framework (Figure 5), viz, *FastMIM*, for faster training speed and easier deployment of AI applications. Inspired by Sim-MIM (Xie et al., 2022), which retains all input tokens during pre-training stage, we directly mask the raw RGB input and keep the illuminated input the same as in the supervised learning producer. This presents a fresh opportunity for *FastMIM* to serve as a ***generic*** framework because no modification is made to the architecture and the input shape. Yet, standard input images of size $224 \times 224$ are inherently used in pre-training stage in common practice (Bao et al., 2021; He et al., 2021). For example, the encoder of ViT-B (Dosovitskiy et al., 2020) needs to tackle 196 input patches in Sim-MIM (Xie et al., 2022). To alleviate memory consumption, we propose a straightforward approach of ***reducing the input resolution***, *e.g.*, from $224 \times 224$ to $128 \times 128$, and the number of input patches is reduced to 64 accordingly, as shown in Figure 2 and 5. We further leverage the HOG target (Dalal & Triggs, 2005; Wei et al., 2022) to compensate for the loss of texture information resulting from the reduction of image resolution. Our main contributions can be summarized as:

- We investigate various configurations of MIM, identify the key design to expedite the pre-training stage and reduce the memory consumption: directly ***reducing the input resolution*** for MIM.
- We elaborate the characteristic of the HOG feature, which is ***almost invariant to the geometric changes in images***. Compared with pixel target, reconstructing HOG target can better compensate for the loss of texture information resulting from the reduction of image resolution.
- Based on the above observations, we propose *FastMIM*, which can expedite the overall pre-training speed by $5\times$ and reduce the memory consumption simultaneously. Extensive experiments demonstrate the effectiveness and efficiency of our proposed framework.

Overall, the heavy memory consumption of previous self-supervised learning frameworks erects an unfortunate barrier for more researchers to dive into this filed. We hope our findings and *FastMIM* can provide avenues and insights for making MIM more accessible to the vision community.

## 2 RELATED WORK

**Masked Image Modeling.** Motivated by tremendously successful BERT (Devlin et al., 2018) and its variants (Brown et al., 2020) for MLM in NLP field, masked image modeling (MIM) is first studied in BEiT (Bao et al., 2021) to pre-train vision transformers (Dosovitskiy et al., 2020). BEiT randomly masks a portion of image patches, and adopts a VQ-VAE (Van Den Oord et al., 2017) as

| ep.\inp. | $64^2$ | $96^2$ | $128^2$ | $160^2$ | $192^2$ | $224^2$ | ep.\inp. | $64^2$ | $96^2$ | $128^2$ | $160^2$ | $192^2$ | $224^2$ |
|---|---|---|---|---|---|---|---|---|---|---|---|---|---|
| 200 | 81.89 | 82.44 | 82.72 | 82.96 | 83.03 | 83.12 | 100 | 82.82 | 83.24 | 83.32 | 83.38 | 83.46 | 83.58 |
| 400 | 82.26 | 82.98 | 83.19 | 83.28 | 83.40 | 83.51 | 400 | 83.07 | 83.51 | 83.76 | 83.78 | 83.85 | 83.93 |
| 800 | 82.85 | 83.22 | 83.51 | 83.59 | 83.68 | 83.79 | 800 | 83.23 | 83.60 | 83.84 | 83.90 | 83.96 | 84.03 |

(a) **Pre-training epoch and input resolution** for ViT-B (left) and Swin-B (right). Normalized raw pixel is used as the prediction target.

| case | inp. | encoder | top-1 | AP$^{\mathbf{b}}$ | AP$^{\mathbf{m}}$ | encoder | top-1 | AP$^{\mathbf{b}}$ | AP$^{\mathbf{m}}$ |
|---|---|---|---|---|---|---|---|---|---|
| pixel | $224^2$ | ViT-B | 83.8 | 50.4 | 45.0 | Swin-B | 84.0 | 52.3 | 46.0 |
| HOG | $224^2$ | ViT-B | 83.9 | 50.9 | 45.2 | Swin-B | 84.2 | 52.5 | 46.3 |
| HOG | $128^2$ | ViT-B | 83.8 | 50.7 | 45.1 | Swin-B | 84.1 | 52.2 | 46.1 |

| encoder | depth | top-1 | encoder | depth | top-1 |
|---|---|---|---|---|---|
| ViT-B | 8 | 82.9 | Swin-B | 22 | 83.9 |
| ViT-B | 10 | 83.4 | Swin-B | 23 | 84.1 |
| ViT-B | 12 | 83.5 | Swin-B | 24 | 84.1 |

(b) **Prediction target and input resolution**.  (c) **Encoder depth** with $128^2$ input.

Table 1: Ablation on input resolution, reconstruction target, and encoder depth in pre-training stage. a) pre-training epoch and input resolution on ViT-B/Swin-B; b) prediction target; c) encoder depth. ImageNet-1K top-1 accuracy, COCO box AP$^{\mathbf{b}}$ and mask AP$^{\mathbf{m}}$ are reported. Default settings are marked in gray.

the visual tokenizer to generate reconstruction targets to finally predict the visual tokens which are corresponding to the masked regions. Recently, several works (Xie et al., 2022; Zhou et al., 2021; Dong et al., 2021; He et al., 2021; Huang et al., 2022; Chen et al., 2022b; Wei et al., 2022; Fang et al., 2022) have revisited MIM as a promising solution to visual representation learning. MAE (He et al., 2021) develops an asymmetric encoder-decoder architecture, with an encoder that operates only on the visible patches (discarding the masked patches), along with a lightweight decoder that reconstructs the masked patches. However, MAE can only be applied to isotropic backbones. In contrast, SimMIM (Xie et al., 2022) proposes to retain all input patches (Dong et al., 2021; Bao et al., 2021; Zhou et al., 2021; Chen et al., 2022a) and thus can serve as generic MIM approach for hierarchical backbones. However, the large amount of input patches not only slow down its pre-training speed, but also incur heavy memory consumption, making SimMIM hard to be deployed on single deep learning machine.

**Expedite MIM.** An obstacle for practical applications of above MIM is the heavy computational cost and long pre-training time. Towards this, UM-MAE (Li et al., 2022) designs a secondary masking strategy to preserve equivalent elements across multiple local window. LoMaR (Chen et al., 2022a) performs masked reconstruction within a small window of $7 \times 7$ patches. GreenMIM (Huang et al., 2022) proposes a group window attention exclusively for hierarchical Swin Transformer (Liu et al., 2021). In contrast to them, our *FastMIM* directly reduce the input resolution, introducing no additional modification to encoder compared with supervised training paradigm, and achieves better trade-off between pre-training speed and fine-tuning accuracy.

**Reconstruction Target in MIM.** In addition to discrete tokens (Bao et al., 2021; Dong et al., 2021) mentioned above, there are still various target signals designed for MIM, such as normalized pixels (He et al., 2021; Xie et al., 2022), HOG (Wei et al., 2022), and latent features (Zhou et al., 2021; Baevski et al., 2022). Among them, pixel and HOG can be directly obtained from original input without extra trained networks. The histogram of oriented gradients (HOG) is a feature descriptor that counts occurrences of gradient orientation in localized portions of an image. And we demonstrate that HOG target is more invariant to the geometric changes in input image and preserves better performance (together with lower pre-training loss) under low resolution input compared to the pixel target.

## 3 REVISIT MASKED IMAGE MODELING

Our *FastMIM* is a simple and straightforward framework based on masked image modeling, which masks a portion of original images, and predicts the masked regions. We start by revisiting MIM by investigating how resolution/target/encoder depth influence the MIM. Preliminaries about MIM and our implementation details are provided in supplementary material.

**Input resolution.** Table 1a explores the impact of pre-training epoch and input resolution on the fine-tuning result of MIM (the result of MAE can be found in supplementary material). It reveals that *a broad range of input resolutions* (*e.g.*, $128^2 \sim 224^2$) perform equally well.

The largest input resolution achieves the best top-1 accuracy as expected. Notably, reducing the input resolution to $128^2$ for Swin-B leads to only a minor decrease of 0.26%/0.17%/0.19% in the final fine-tuning results for 100/400/800 epochs, respectively. When the input resolution is set to $224^2$, the encoder has to handle a substantial number of image patches

| Model | HOG Target | | Pixel Target | |
|---|---|---|---|---|
| | Input | Loss | Input | Loss |
| ViT-B | $224^2/128^2$ | 0.028/0.031 | $224^2/128^2$ | 0.408/0.494 |
| Swin-B | $192^2/128^2$ | 0.034/0.037 | $192^2/128^2$ | 0.521/0.619 |
| Swin-L | $192^2/128^2$ | 0.031/0.035 | $192^2/128^2$ | 0.514/0.594 |

Table 2: Ablation on value of pre-training loss. ViT-B is trained with 800 epochs, and Swin-B/L are trained with 400 epochs.

(*e.g.*, $N_e$=$56^2$ in Swin-B stage-1), leading to a heavy memory burden and a long computing time. In contrast, setting the input resolution to $128^2$ naturally reduces the number of image patches to $N_e$=$32^2$, which is 70% less than the $224^2$ input, while maintaining similar performance. However, further reducing the input resolution results in a significant drop in fine-tuning top-1 accuracy, likely because lower resolution and fewer input patches discard too much essential information, which is indispensable during the reconstruction stage.

**Prediction target.** Table 1b compares the effects of two prediction targets. The most straightforward target involves predicting the colors of original pixels. Specifically, we use normalized RGB values following (He et al., 2021; Xie et al., 2022). Histograms of Oriented Gradients (HOG) (Dalal & Triggs, 2005; Wei et al., 2022) is a feature descriptor that counts occurrences of gradient orientation in localized portions of an image. Here we minimize the $\ell_2$ distance between the model's prediction and the ground-truth RGB value/HOG feature. Under the setting of MIM pre-training and $224^2$ input, both prediction targets exhibit similar performance on both classification and detection tasks. But when reducing the input resolution to $128^2$, pixel and HOG exhibit distinct characteristics, which will be further analyzed later.

**Encoder depth in pre-training.** As mentioned above, reducing the input resolution (encoder patches) can help ease memory overhead and save training time. Additionally, there is another straightforward method to save computational cost: reducing the number of parameters (encoder depth) trained in the pre-training stage. Inspired by the layer decay strategy (where shallow layer has a smaller learning rate compared to deep layer) in fine-tuning of BEiT (Bao et al., 2021), we conjecture that shallow layers are more important than deep layers during the pre-training phase. Table 1c illustrates that discarding the last several layers (blocks) in pre-training (discarded layers will be re-initialized in fine-tuning) yields almost the same performance compared to the original setting (the third row in Table 1c). Note that the hierarchical Swin-B encoder comprises four stages (*e.g.*, [2,2,18,2]), Table 1c only presents the results of [2,2,18,0] (the first row) and [2,2,18,1] (the second row). If we discard layers in the third stage, *e.g.*, [2,2,16,2], the fine-tuning performance will drop to 83.4%. This phenomenon has also been observed in prior work (Huang et al., 2023). Our study further offers additional validation across both the isotropic ViT and the hierarchical Swin.

**Discussion on Epoch/Resolution/Target.** Figure 4 further presents the result of utilizing both HOG and pixel targets. It is noteworthy that HOG demonstrates superior stability and delivers better performance as the resolution is reduced. The accuracy improves consistently as training epochs increase. We observe that HOG target begins to saturate at 800/400 epochs for ViT-B/Swin-B, in contrast to the pixel target. One main reason is that the HOG is more resilient to ambiguity by histogramming local gradients (Wei et al., 2022). Besides, HOG can maintain better performance when the input resolution is reduced due to its characteristic. To elaborate it, we first qualitatively compare HOG to pixel as the prediction target in Figure 3. While reducing the image resolution can significantly expedite the training process, crucial information such as detailed textures and edges will be lost when using the pixel target. However, HOG is more resistant to resolution changes,

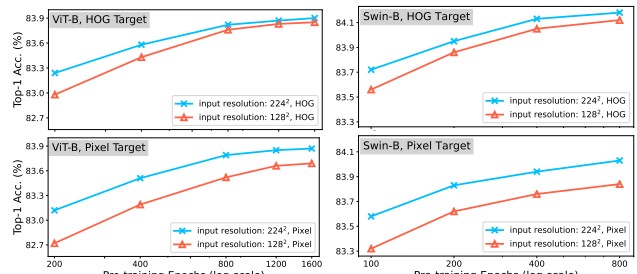

Figure 4: Ablation on epoch/resolution/target. HOG achieves better result with low-resolution input compared with the raw pixel.

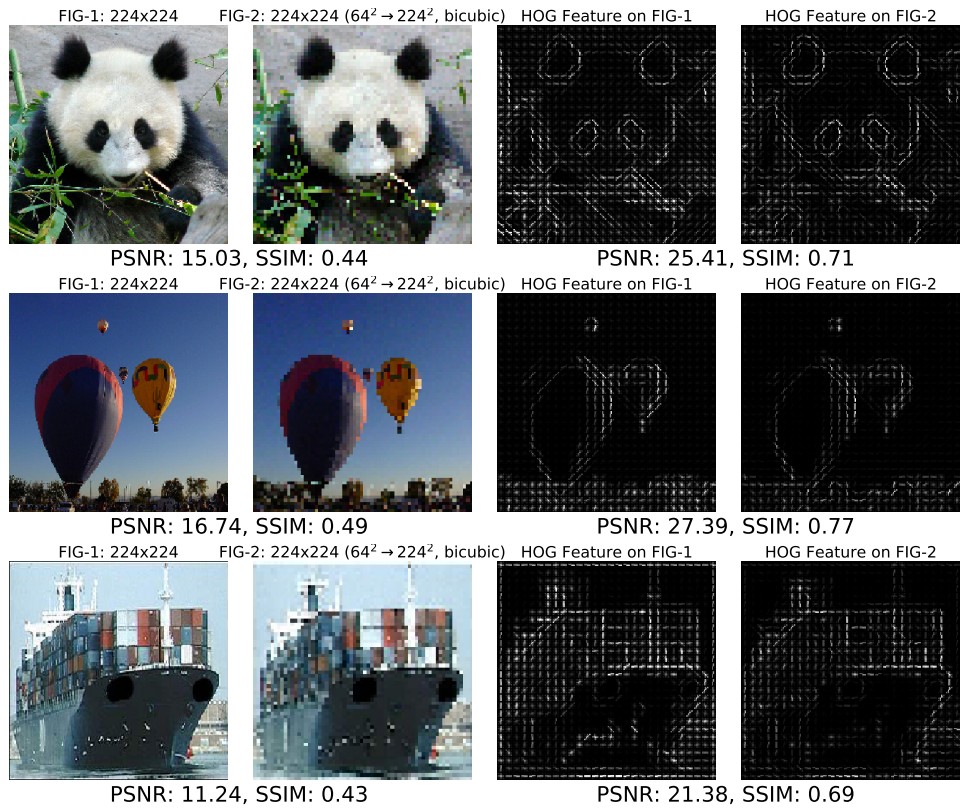

Figure 3: Visualization on pixel target and HOG target. We choose PSNR (dB) and SSIM to evaluate the similarity between two images (features). HOG target can preserve better texture information under low resolution input compared to pixel target.

making it ideal for our *FastMIM*. Furthermore, we also display the values of pre-training loss in Table 2. It is obvious that HOG can reduce the gap between loss values of different resolutions. Moreover, the loss of using the HOG target is significantly lower than that of using the pixel target, demonstrating that HOG can effectively mitigate the risk of ambiguity during reconstruction in MIM.

# 4 APPROACH

Our *FastMIM* pre-trains vision backbones through masked image modeling, and is also a reinforced version of SimMIM (Xie et al., 2022), as illustrated in Figure 5. In principle, it is straightforward and convenient to replace the encoder with other vision backbones in aforementioned MIM pre-training framework. We choose the representative isotropic and hierarchical vision transformers, *i.e.*, ViT (Dosovitskiy et al., 2020) and Swin Transformer (Liu et al., 2021) as our baselines. We directly mask the input image (*e.g.*, $\mathbf{X} \in \mathbb{R}^{128 \times 128 \times 3}$) with the mask token (*e.g.*, learnable vector [MASK] $\in \mathbb{R}^{1 \times 1 \times 3}$). The ViT encoder embeds patches by a linear projection added with positional embeddings (PE), while there is no extra PE for the decoder. As for Swin, the window size for Swin-B and Swin-L is set to 7 and 14 following (Xie et al., 2022; Huang et al., 2022), respectively.

## 4.1 *FastMIM* FRAMEWORK

**Masked Input.** The input image is randomly cropped and resized to **128×128**. Therefore, the number of patches (pixels) is reduced to $N_e$=64 and $N_e$=$32^2/16^2/8^2/4^2$ for ViT and Swin, receptively. We leverage a per-sample random mask strategy, and set the mask size to the same value as the last layer's patch size of the encoder. Specifically, the mask size is 16×16 and 32×32 for ViT and Swin, respectively. The mask ratio is set to 0.75 according to ablation study in supplementary material.

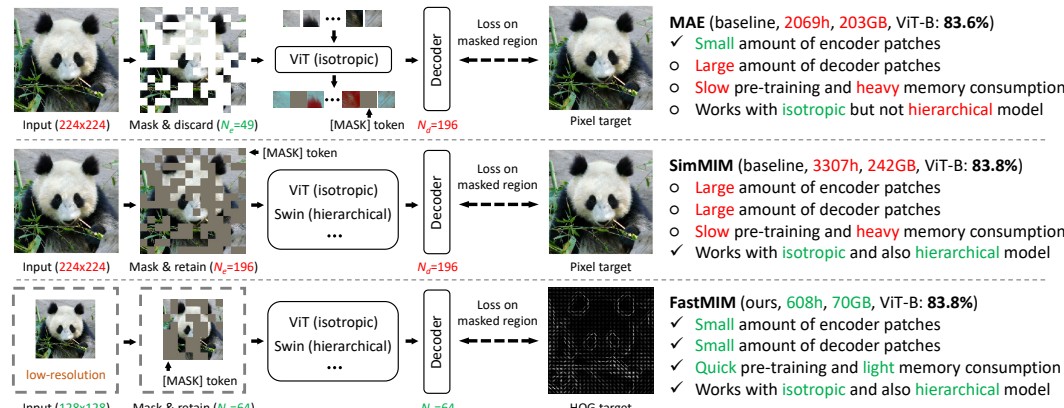

Figure 5: Comparison among MAE (He et al., 2021), SimMIM (Xie et al., 2022) and our *FastMIM*. MAE randomly masks and discards input patches, limiting its use to pre-training isotropic ViT which generates single-scale intermediate features. SimMIM preserves all patches and can serve as a generic framework for various backbones, but requires processing a large number of patches. In contrast, *FastMIM* reduces input resolution and uses HOG target, resulting in a simpler and more efficient approach. *FastMIM* (i) pre-train faster; (ii) has lower memory consumption; (iii) can serve as a generic framework for different architectures; and (iv) achieves comparable or better performance than previous methods.

**Decoder.** The decoder is only used in pre-training stage to perform the reconstruction task. Note that the input resolution of *FastMIM* is set to $128 \times 128$, the encoder outputs for ViT-L and Swin-L are of size $64 \times 1024$ and $16 \times 1536$, respectively. The memory usage of our decoder is indeed 65% less than that of MAE (He et al., 2021). According to the ablation study in supplementary material, the decoder sizes for ViT-B/ViT-L/Swin-B/Swin-L are set to 1b256d/8b512d/4b256d/4b512d, respectively.

**Prediction Target.** We choose Histograms of Oriented Gradients (HOG) (Dalal & Triggs, 2005) features as the target following MaskFeat (Wei et al., 2022). We first obtain an entire HOG feature on the whole image and then minimize the $\ell_2$ distance between the output of *FastMIM* and original HOG feature on masked region. The number of orientation bins is set to 9, and spatial cell is set to $8 \times 8$. Discussion on HOG is shown in Sec. 3

### 4.2 *FastMIM-P*: PROGRESSIVELY ENLARGE THE INPUT

To further improve the scalability of *FastMIM*, we propose to progressively enlarge the input resolution during the pre-training stage, viz, *FastMIM-P*. Although HOG can preserve the texture information when reducing the input resolution to some extent, the performance of model with high capacity, *e.g.*, Swin-L, still has small gap (-0.3% in Table 3) compared to the counterpart trained with high-resolution input images. More specifically, in contrast to *FastMIM* that trains Swin-L in fixed $128^2$ inputs, *FastMIM-P* trains Swin-L in $128^2/160^2/192^2$ for 200/100/100 epochs (400 epochs in total), and achieve better trade-off between accuracy and training time, as shown in the last row of Table 3. Additionally, we report the results of *FastMIM-P* on ViT-L, trained with input resolutions of $128^2/160^2/192^2$ for 500/200/100 epochs respectively. Implementation details can be found in the supplementary material.

**Discussion.** As shown in Table 3, *FastMIM-P* achieves better performance compared to *FastMIM* with less pre-training time. However, as the input resolution continually increases, the GPU memory consumption will inevitably increase. The resolution and training schedule need to be carefully designed to achieve a better space-time trade-off. In addition, we find that reducing the input resolution results in slightly more pronounced accuracy degradation as the model scales up from ViT-B/Swin-B to ViT-L/Swin-L. This validates that larger models indeed tend to be more data-hungry, and a smaller number of input tokens may lead to insufficient pretraining. However, by leveraging the progressively enlarged input resolution strategy, we effectively mitigate this issue and achieve significant improvements in accuracy for larger models.

| Framework | Model | # Params | PT Ep. | Hours/Ep. | PT Hours | FT Ep. | Top-1 (%) |
|---|---|---|---|---|---|---|---|
| *Supervised pre-training* | | | | | | | |
| Training from scratch in MAE | ViT-B | 86M | 0 | $1.6^{\dagger,\ddagger}$ | $490^{\dagger,\ddagger}$ | 300 | 82.3 |
| Training from scratch in Swin | Swin-B | 88M | 0 | $2.5^{\dagger,\ddagger}$ | $744^{\dagger,\ddagger}$ | 300 | 83.5 |
| PT (192) then FT (224) in SimMIM | Swin-L | 197M | 300 | $3.8^{\ddagger}$ | $1139^{\ddagger}$ | 100 | 83.5 |
| *Self-supervised pre-training with contrastive learning* | | | | | | | |
| MoCov3 | ViT-B | 86M | 800 | - | - | 100 | 83.2 |
| DINO | ViT-B | 86M | 800 | - | - | 100 | 82.8 |
| *Self-supervised pre-training with masked image modeling on isotropic ViT* | | | | | | | |
| BEiT | ViT-B | 86M | 800 | 2.4 | 1920 | 100 | 83.2 |
| MAE | ViT-B | 86M | 1600 | 1.3 | 2069 | 100 | 83.6 |
| MAE | ViT-L | 307M | 1600 | 2.0 | 3260 | 100 | 85.9 |
| SimMIM | ViT-B | 86M | 800 | 4.1 | 3307 | 100 | 83.8 |
| LoMaR | ViT-B | 86M | 800 | 1.4 | 1120 | 100 | 83.8 |
| CAE | ViT-B | 86M | 1600 | - | - | 100 | 83.9 |
| MaskFeat | ViT-B | 86M | 800 | $1.6^{\ddagger}$ | $1264^{\ddagger}$ | 100 | 84.0 |
| iBOT | ViT-B | 86M | 1600 | - | - | 100 | 84.0 |
| PeCo | ViT-B | 86M | 800 | - | - | 100 | 84.5 |
| FastMIM (ours) | ViT-B | 86M | 400 | 0.8 | 304 | 100 | 83.6 |
| FastMIM (ours) | ViT-B | 86M | 800 | 0.8 | 608 | 100 | 83.8 |
| FastMIM (ours) | ViT-L | 307M | 800 | 1.3 | 1062 | 100 | 85.1 |
| FastMIM-P (ours) | ViT-L | 307M | 800 | 1.8 | 1434 | 100 | 85.7 |
| *Self-supervised pre-training with masked image modeling on hierarchical Swin* | | | | | | | |
| SimMIM | Swin-B | 88M | 800 | 2.0 | 1609 | 100 | 84.0 |
| GreenMIM | Swin-B | 88M | 800 | 1.1 | 887 | 100 | 83.8 |
| FastMIM (ours) | Swin-B | 88M | 400 | 0.8 | 336 | 100 | 84.1 |
| *Self-supervised pre-training with masked image modeling on hierarchical Swin* | | | | | | | |
| SimMIM | Swin-L | 197M | 800 | 3.5 | 2821 | 100 | $85.5^{\ddagger}$ |
| GreenMIM | Swin-L | 197M | 800 | 1.3 | 1067 | 100 | 85.1 |
| FastMIM (ours) | Swin-L | 197M | 400 | 1.4 | 544 | 100 | 85.2 |
| FastMIM (ours) | Swin-L | 197M | 800 | 1.4 | 1088 | 100 | 85.4 |
| FastMIM-P (ours) | Swin-L | 197M | 400 | 1.8 | 736 | 100 | 85.5 |

Table 3: Comparison with state-of-the-art MIM methods. "PT Ep." refers to the number of pre-training epoch. "Hours/Ep." refers to GPU hours per epoch. "PT Hours" refers to total pre-training GPU hour. "FT Ep." refers to fine-tuning epoch. We report top-1 accuracy on ImageNet-1K validation set with the ViT-B/Swin-B/Swin-L models. ($\dagger$: we report the total hours in fine-tuning stage. $\ddagger$: result tested by us.)

## 5 EXPERIMENTS

### 5.1 IMAGENET-1K CLASSIFICATION

Table 3 reports the top-1 accuracy on ImageNet validation set (Deng et al., 2009). We compare our *FastMIM* with vision transformers trained via supervised pre-training, self-supervised training with contrastive learning, and self-supervised training with MIM. Compared to the models trained from scratch with random initialization, we find that pre-training through *FastMIM* significantly improves performances on both ViT-B and Swin-B by +1.5% and +0.6%, respectively. Notably, the total pre-training hour of *FastMIM* based Swin-B is 336, the total pre-training and fine-tuning time is 584 hours, which is *less* than the 744 hours for training from scratch. This result indicates that our *FastMIM* can also serve as a regular training paradigm for classification, and is more efficient and effective than the commonly used scheme. Besides, ViT-B pre-trained with 400 epochs via *FastMIM* achieves 83.6% top-1 accuracy on ImageNet-1K, which is +1.3% higher than the baseline counterpart. And the total pre-training and fine-tuning time is 467 hours, which is also *less* than 490 hours spent by conventional supervised scheme. These improvements suggest that our *FastMIM* can effectively expedite the pre-training process for various vision backbones.

Moreover, we compare *FastMIM* with previous state-of-the-art self-supervised methods for isotropic ViT-B, such as BEiT (Bao et al., 2021), MAE (He et al., 2021), SimMIM (Xie et al., 2022), CAE (Chen et al., 2022b), MaskFeat (Wei et al., 2022), iBOT (Zhou et al., 2021), and PeCo (Dong et al., 2021). Among them, CAE (Chen et al., 2022b) uses extra 250M DALL-E data (Ramesh et al., 2021) to pre-train the tokenizer, iBOT (Zhou et al., 2021) uses an extra momentum ViT as the online

| Framework | Model | Param | PTE | Hrs/Ep. | PT Hrs | FTE | FT Hrs | Total Hrs | Top-1 (%) |
|---|---|---|---|---|---|---|---|---|---|
| Supervised (He et al., 2021) | ViT-B | 86M | - | 1.6 | - | 300 | 490 | 490 | 82.3 |
| FastMIM (ours) | ViT-B | 86M | 800 | 0.8 | 608 | 100 | 163 | 771 | 83.8 (+1.5) |
| Supervised (Liu et al., 2021) | Swin-B | 88M | - | 2.5 | - | 300 | 744 | 744 | 83.5 |
| FastMIM (ours) | Swin-B | 88M | 400 | 0.8 | 336 | 100 | 248 | 584 | 84.1 (+0.6) |
| Supervised (Xie et al., 2022) | Swin-L | 197M | 300 | 2.6 | 780 | 100 | 359 | 1139 | 83.5 |
| FastMIM (ours) | Swin-L | 197M | 800 | 1.4 | 1088 | 100 | 359 | 1447 | 85.4 (+0.9) |
| Supervised (Chu et al., 2021) | Twins-L | 99M | - | 2.8 | - | 300 | 832 | 832 | 83.7 |
| FastMIM (ours) | Twins-L | 99M | 800 | 0.9 | 716 | 100 | 277 | 993 | 84.0 (+0.3) |
| Supervised (Wang et al., 2021) | PVTv1-L | 61M | - | 2.1 | - | 300 | 624 | 624 | 81.7 |
| FastMIM (ours) | PVTv1-L | 61M | 800 | 0.7 | 592 | 100 | 208 | 800 | 82.9 (+1.2) |
| Supervised (Wang et al., 2022) | PVTv2-B2 | 25M | - | 1.7 | - | 300 | 504 | 504 | 82.0 |
| FastMIM (ours) | PVTv2-B2 | 25M | 800 | 0.6 | 448 | 200 | 336 | 784 | 82.6 (+0.6) |
| Supervised (Wang et al., 2022) | PVTv2-B5 | 82M | - | 3.8 | - | 300 | 1152 | 1152 | 83.8 |
| FastMIM (ours) | PVTv2-B5 | 82M | 800 | 1.3 | 1026 | 200 | 768 | 1794 | 84.3 (+0.5) |
| Supervised (Liu et al., 2022) | ConvNeXt-T | 28M | - | 1.4 | - | 300 | 415 | 415 | 82.1 |
| FastMIM (ours) | ConvNeXt-T | 28M | 800 | 0.5 | 444 | 300 | 415 | 859 | 82.6 (+0.5) |
| Supervised (Liu et al., 2022) | ConvNeXt-B | 89M | - | 2.7 | - | 300 | 816 | 816 | 83.8 |
| FastMIM (ours) | ConvNeXt-B | 89M | 800 | 1.0 | 776 | 300 | 816 | 1592 | 84.0 (+0.2) |
| Supervised (Woo et al., 2023) | ConvNeXt V2-B | 89M | - | 2.7 | - | 300 | 824 | 824 | 84.3 |
| FCMAE (Woo et al., 2023) | ConvNeXt V2-B | 89M | 800 | 2.0 | 1592 | 100 | 275 | 1867 | 84.6 (+0.3) |
| FCMAE (Woo et al., 2023) | ConvNeXt V2-B | 89M | 800 | 2.0 | 1592 | 300 | 824 | 2416 | 84.7 (+0.4) |
| FastMIM (ours) | ConvNeXt V2-B | 89M | 800 | 1.2 | 948 | 300 | 824 | 1772 | 84.6 (+0.3) |
| Supervised (Guo et al., 2022) | CMT-S | 25M | - | 2.8 | - | 300 | 840 | 840 | 83.5 |
| FastMIM (ours) | CMT-S | 25M | 800 | 1.0 | 768 | 200 | 560 | 1328 | 83.9 (+0.4) |
| Supervised (Guo et al., 2022) | CMT-B | 46M | - | 6.4 | - | 300 | 1925 | 1925 | 84.5 |
| FastMIM (ours) | CMT-B | 46M | 800 | 1.5 | 1236 | 200 | 1283 | 2519 | 85.0 (+0.5) |

Table 4: Comparison with supervised training method on more backbones. "PTE" is the number of pre-training epoch. "Hrs/Ep." means GPU hours per epoch. "PT Hrs" is total pre-training GPU hour. "FTE" is fine-tuning epoch. We report top-1 accuracy on ImageNet-1K validation set. "FT Hrs" is total fine-tuning GPU hour. "Total Hrs" is the total training time.

tokenizer, and PeCo (Dong et al., 2021) leverages both VQ-VAE (Van Den Oord et al., 2017) tokenizer and MoCov3 (Chen et al., 2021) framework. These extra modules introduce non-negligible memory overhead and considerably longer training time. MAE (He et al., 2021), SimMIM (Xie et al., 2022) and MaskFeat (Wei et al., 2022) are the most comparable methods. Our approach achieves 83.8% top-1 accuracy, which is on par with above MIM frameworks. As for the computational cost, *FastMIM* is $1.6\times/2\times/5.1\times$ faster than MAE/SimMIM/ MaskFeat, and reduces the GPU memory consumption by 50%~70% when compared to MAE and SimMIM, as shown in Figure 2. We also evaluate *FastMIM* with hierarchical Swin Transformer (Liu et al., 2021). Our approach obtains 84.1% top-1 accuracy with the Swin-B backbone, which is superior to the supervised learning counterpart. When compared to the recently proposed GreenMIM (Huang et al., 2022), which exclusively designs a group window attention for pre-training Swin, *FastMIM* achieves slightly better result (+0.3%) with only half of the pre-training time, and less memory usage (108.3 *vs.* 121.6). As for Swin-L, we fine-tune the SimMIM (Xie et al., 2022) through the grid search, where the result is slightly better than that reported in SimMIM paper. When pre-trained with 400 epochs, *FastMIM* achieves 85.2% top-1 accuracy and surpasses the 800 epochs GreenMIM (Huang et al., 2022). When the pre-training epoch is extend to 800, *FastMIM* further improves the Swin-L by +0.2%. Besides, *FastMIM-P* achieves 85.5% top-1 accuracy, which is at-par with the result obtained by SimMIM trained with 800 epochs, while our pre-training speed is $\sim4\times$ faster. The corresponding results demonstrate the effectiveness and efficiency of our method, especially the substantial improvements on pre-training speed and memory consumption over previous MIM frameworks.

## 5.2 MORE RESULTS ON VARIOUS BACKBONES

Our *FastMIM* can serve as a generic MIM framework for various vision backbones, including vanilla isotropic ViT (Dosovitskiy et al., 2020), hierarchical Swin Transformer (Liu et al., 2021), Twins (Chu et al., 2021), PVT (Wang et al., 2021; 2022), ConvNeXt (Liu et al., 2022), and CMT (Guo et al., 2022). We conduct experiments based on above vision backbones and

| Framework | Backbone | IN-1K FT | FT Epoch | Object Detection | | | Instance Segmentation | | |
|---|---|---|---|---|---|---|---|---|---|
| | | | | $AP^b$ | $AP^b_{50}$ | $AP^b_{75}$ | $AP^m$ | $AP^m_{50}$ | $AP^m_{75}$ |
| *Training from scratch (random initialization)* | | | | | | | | | |
| Benchmarking | ViT-B | ✗ | 0 | 48.9 | - | - | 43.6 | - | - |
| *Self-supervised pre-training, follow the coco fine-tuning setup in MAE* | | | | | | | | | |
| MAE | ViT-B | ✗ | 1600 | 48.1 | 69.3 | 53.3 | 43.2 | 66.3 | 46.7 |
| FastMIM (ours) | ViT-B | ✗ | 800 | 48.6 | 70.5 | 53.6 | 43.5 | 67.0 | 46.9 |
| BEiT | ViT-B | ✗ | 800 | 49.8 | - | - | 44.4 | - | - |
| MAE | ViT-B | ✗ | 1600 | 50.3 | 70.9 | 55.6 | 44.9 | 68.3 | 49.0 |
| FastMIM (ours) | ViT-B | ✗ | 800 | 50.7 | 71.3 | 56.0 | 45.1 | 68.6 | 49.3 |
| *Self-supervised pre-training, follow the coco fine-tuning setup in GreenMIM* | | | | | | | | | |
| SimMIM | Swin-B | ✗ | 800 | 50.4 | 70.9 | 55.5 | 44.4 | 68.2 | 47.9 |
| GreenMIM | Swin-B | ✗ | 800 | 50.0 | 70.7 | 55.4 | 44.1 | 67.9 | 47.5 |
| FastMIM (ours) | Swin-B | ✗ | 400 | 50.3 | 71.0 | 55.3 | 44.4 | 68.2 | 48.0 |
| *Self-supervised pre-training, follow the coco fine-tuning setup in SimMIM* | | | | | | | | | |
| SimMIM[†] | Swin-B | ✓ | 800 | 52.3 | 73.4 | 57.9 | 46.1 | 70.6 | 50.2 |
| FastMIM (ours) | Swin-B | ✗ | 400 | 51.9 | 72.9 | 57.2 | 45.8 | 70.2 | 49.5 |
| FastMIM (ours) | Swin-B | ✓ | 400 | 52.2 | 73.3 | 57.6 | 46.1 | 70.4 | 50.2 |
| SimMIM[†] | Swin-L | ✓ | 800 | 53.7 | 74.8 | 58.6 | 47.2 | 71.9 | 51.5 |
| FastMIM (ours) | Swin-L | ✓ | 400 | 53.2 | 74.4 | 58.1 | 46.9 | 71.6 | 51.3 |
| FastMIM-P (ours) | Swin-L | ✓ | 400 | 53.6 | 74.9 | 58.4 | 47.2 | 72.0 | 51.5 |

Table 5: COCO object detection and instance segmentation. All methods are based on the Mask R-CNN (He et al., 2017) architecture with the FPN neck. "IN-1K FT" indicates whether use the model fine-tuned on ImageNet-1K for the initialization on COCO. ([†]: our implementation, the IN-1K fine-tuned checkpoint is downloaded from github, and the final $AP^b$ is similar with the number reported in SimMIM.)

compare the top-1 accuracy with previous supervised training results. As shown in Table 4, our *FastMIM* consumes fewer pre-training hours but obtains consistently better performance on all architectures. In particular, our *FastMIM* achieves 82.9/82.6/84.3/82.6/83.9/85.0 top-1 accuracy with PVTv1-L/PVTv2-B2/PVTv2-B5/ConvNeXt-T/CMT-S/CMT-B, which is +1.2/+0.6/+0.5/ +0.6/+0.4/+0.5 better than the supervised training counterparts. These results demonstrate the efficiency and effectiveness of our generic *FastMIM* framework.

## 5.3 OBJECT DETECTION AND INSTANCE SEGMENTATION

We show the transfer learning results on COCO (Lin et al., 2014) in Table 5. We first follow the fine-tuning setting in MAE (He et al., 2021; Li et al., 2021), and report results of two considered training lengths: 25 and 100 epochs. Our *FastMIM* yields up to 0.5 and 0.4 higher $AP^{box}$ than MAE in both settings, and the pre-training hours is much less than MAE. Then we directly use the code base of the Green-MIM (Huang et al., 2022) without any modification to the fine-tuning strategy. Compared with the Swin-B pre-trained by GreenMIM, our approach performs prominently better in terms of all met-

| Framework | Backbone | PT Epoch | PT Hours | mIoU |
|---|---|---|---|---|
| *Self-supervised pre-training, follow the setup in MAE* | | | | |
| MoCov3 | ViT-B | - | - | 47.3 |
| BEiT (w/ DALL-E) | ViT-B | 800 | 1920 | 47.1 |
| MAE | ViT-B | 1600 | 2069 | 48.1 |
| PeCo | ViT-B | 800 | - | 48.5 |
| CAE (w/ DALL-E) | ViT-B | 800 | - | 49.7 |
| FastMIM (ours) | ViT-B | 800 | 608 | 49.4 |
| *Self-supervised pre-training, follow the setup in SimMIM* | | | | |
| SimMIM | Swin-B | 800 | 1609 | 52.8 |
| FastMIM (ours) | Swin-B | 400 | 336 | 52.6 |

Table 6: Semantic segmentation on ADE20K. We report the results of ViT-B and Swin-B following two settings.

rics, *e.g.*, +0.3% improvement in both $AP^{box}$ and $AP^{mask}$, with less pre-training epochs (-400). Besides, our approach still obtains similar results with the SimMIM (Xie et al., 2022). Our *FastMIM* can also scale up to larger models and obtain better performance. We conduct the experiments by following the settings in SimMIM, and use their public checkpoints for direct comparisons. Our *FastMIM* achieves 52.2 and 53.2 $AP^{box}$ (46.1 and 46.9 $AP^{mask}$) for Swin-B and Swin-L, respectively, which are comparable to the SimMIM, and are achieved with much less pre-training cost. Furthermore, *FastMIM-P* obtains almost the same performance as SimMIM with faster pre-training speed. In general, the masked image modeling based methods show the potential to substantially

improve detection transfer learning results, and our *FastMIM* can save a lot of pre-training overhead and bring impressive pre-training efficiency.

## 5.4 ADE20K SEMANTIC SEGMENTATION

Table 6 presents the result of *FastMIM* on ADE20K (Zhou et al., 2017). Following the setup in He et al. (2021), we achieve 49.4 mIoU, +1.3 better than MAE while requiring only 30% of its pre-training time. We note that the performance is also comparable to CAE (Chen et al., 2022b), which leverages extra DALL-E data to pre-train its tokenizer. Besides, we follow the setup in SimMIM (Xie et al., 2022) and obtain 52.6 mIoU, which is also comparable to the 52.8 obtained by SimMIM.

| Model | Param | Pixel Target | | HOG Target | |
|---|---|---|---|---|---|
| | | PT inp. | Top-1 (%) | PT inp. | Top-1 (%) |
| ViT-B | 86M | $224^2/128^2$ | 83.8/83.6(-0.2) | $224^2/128^2$ | 83.8/83.8(-0.0) |
| ViT-L | 304M | $224^2/128^2$ | 84.9/84.4(-0.5) | $224^2/128^2$ | 85.1/85.0(-0.1) |
| Swin-B | 88M | $192^2/128^2$ | 84.0/83.8(-0.2) | $192^2/128^2$ | 84.1/84.1(-0.0) |
| Swin-L | 197M | $192^2/128^2$ | 85.5/85.1(-0.4) | $192^2/128^2$ | 85.6/85.4(-0.2) |
| CMT-S | 25M | $224^2/128^2$ | 83.9/83.6(-0.3) | $224^2/128^2$ | 84.0/83.9(-0.1) |
| CMT-B | 46M | $224^2/128^2$ | 85.0/84.6(-0.4) | $224^2/128^2$ | 85.3/85.1(-0.2) |
| PVTv2-b2 | 25M | $224^2/128^2$ | 82.5/82.2(-0.3) | $224^2/128^2$ | 82.7/82.6(-0.1) |
| PVTv2-b5 | 82M | $224^2/128^2$ | 84.3/84.0(-0.3) | $224^2/128^2$ | 84.3/84.3(-0.0) |

Table 7: Ablation on reconstruction target and input resolution for different vision backbones, pre-trained with 800 epochs.

| PT Data | Days | PT Input | FT Input | Top-1 |
|---|---|---|---|---|
| IN-1K | ~1.6 | $128^2$ | $224^2$ | 84.1 |
| IN-1K | ~1.6 | $128^2$ | $384^2$ | 85.3 |
| IN-1K | ~7 | $224^2$ | $384^2$ | 85.4 |
| IN-1K | ~18 | $448^2$ | $224^2$ | 84.3 |
| IN-22K | ~6.5 | $128^2$ | $384^2$ | 86.1 |

Table 8: Swin-B with larger PT/FT resolutions on ImageNet-1K.

## 5.5 ABLATION OF HOG AND RESOLUTION

**Robustness of HOG.** Table 7 presents additional results for two reconstruction targets in relation to changes in input resolution. Notably, the HOG target outperforms the raw pixel target with various encoders by a substantial margin.

**Larger Resolution.** We conduct ablations based on Swin-B with larger pre-training and fine-tuning inputs in Table 8. The performance of Swin-B improves significantly when it is transferred using high-resolution inputs. Moreover, fine-tuning results can be further improved by pre-training the model with more data. Pretraining on larger datasets such as ImageNet-22K can significantly improve performance. However, pre-training the model with larger input may not provide many immediate benefits for classification tasks.

## 5.6 EXTENSION TO OTHER MASKED IMAGE MODELING FRAMEWORK

In this section, we evaluate the effectiveness of *FastMIM* within a distillation-based MIM framework (Bai et al., 2023), where the teacher model is a fine-tuned version. Specifically, to pretrain ViT-B and PVTv2-b2, we leverage their larger homogeneous counterparts, ViT-L and PVTv2-b5, as teachers. The training objective follows the loss function $L = L_{hog} + \sum_i ||\sigma(z_i^S) - z_i^T||_1$. As shown in Table 9, our framework can be seamlessly integrated into other MIM approaches, demonstrating performance improvements even when using low input resolution during pretraining.

| Model | ViT-B | PVTv2-b2 |
|---|---|---|
| Baseline | 82.3 | 82.0 |
| FastMIM | 83.6 | 82.4 |
| FastMIM + KD | 83.8 | 82.6 |

Table 9: Models are pretrained with 400 ep and finetuned with 100 ep.

## 6 CONCLUSION

This paper presents a simple yet effective *FastMIM* to expedite the self-supervised MIM pre-training for various vision backbones. As a generic framework, we directly mask input images, allowing all encoders to be trained in the same way as supervised learning. Besides, simply reducing the image resolution and reconstructing HOG target can train both isotropic and hierarchical architectures $5\times$ faster and save the GPU memory consumption by up to ~70% compared with previous approaches, while obtaining a comparable performance on classification and other downstream vision tasks. We hope our observations and the simple framework can make MIM more practicable and demolish the barrier so that more researchers can dive into this field.

# 7 DISCUSSION ON POTENTIAL LIMITATIONS OF FASTMIM

While our proposed FastMIM approach demonstrates significant improvements in training efficiency and resource utilization, there are inherent limitations to the method. Specifically, while using HOG features effectively compensates for the loss of texture information caused by reduced image resolution, HOG's reduced sensitivity to color and texture variations may make it less suitable for tasks requiring precise appearance modeling. For vision classification and related tasks, where global structure and high-level semantic features are more critical, this approach can yield performance improvements, as demonstrated in our experimental results. However, for tasks that heavily depend on fine-grained visual details, such as image generation or super-resolution, the loss of detailed color information could hinder performance. This limitation underscores the need for caution when applying FastMIM to such specialized tasks. Future work could explore hybrid approaches that integrate HOG with complementary representations to enhance adaptability across a broader range of tasks.

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

# A   APPENDIX

| case | mask loc. | [M] loc. | learn [M] | top-1 |
|------|-----------|----------|-----------|-------|
| MAE | patch | decoder | ✗ / ✓ | 83.2 / 83.6 |
| MIM | patch | encoder | ✗ / ✓ | 83.6 / 83.7 |
| MIM | image | encoder | ✗ / ✓ | 83.6 / 83.7 |

| size\ratio | 0.50 | 0.65 | 0.75 | 0.85 |
|-----------|------|------|------|------|
| 8×8 | 83.3 | 83.4 | 83.3 | 82.9 |
| 16×16 | 83.3 | 83.5 | 83.6 | 83.5 |
| 32×32 | 83.6 | 83.4 | 83.2 | 83.0 |

| size\ratio | 0.50 | 0.65 | 0.75 | 0.85 |
|-----------|------|------|------|------|
| 16×16 | 83.9 | 84.0 | 83.8 | 83.6 |
| 32×32 | 83.9 | 83.9 | 84.1 | 83.8 |
| 64×64$^\dagger$ | 83.6 | 83.1 | - | - |

(a) **Mask strategy.** ViT-B as encoder, raw pixel as prediction target, pre-trained with 800 epochs.

(b) **Mask size and mask ratio** for ViT-B (left) and Swin-B (right). Pre-trained with $128^2$ input, HOG target, and 400 epochs.

Table 10: Ablation studies on ImageNet-1K. a) mask strategies for MAE and MIM; b) mask size and ratio. $^\dagger$: when the mask size is set to 64×64 with $128^2$ input, mask ratio > 0.5 will lead to the same result. Default settings are marked in gray.

## A.1   PRELIMINARY OF MIM

**Notations.** Following commonly used configurations (He et al., 2021; Bao et al., 2021; Xie et al., 2022), given an input image $\mathbf{X} \in \mathbb{R}^{H \times W \times C}$, where $H$, $W$, and $C$ are the height, width, and number of channels, some of the pixels in $\mathbf{X}$ are randomly masked out by being replaced with a mask token, denoted as [M]. Let $\mathbf{S} \in \{0, 1\}^{H \times W \times C}$ denotes the spatial mask, where 0 indicates a pixel[1] is invisible for the encoder, and 1 indicates a pixel is visible.

**Framework.** MIM learns representations by predicting the masked area of an input $\mathbf{X}$. Existing MIM methods can be roughly classified into two categories: (i) MAE (He et al., 2021; Huang et al., 2022) discards the masked area and only the visible part is sent to the encoder for latent feature extracting, then the decoder reconstructs the masked part from latent representation and mask token; (ii) SimMIM (Xie et al., 2022; Bao et al., 2021; Wei et al., 2022; Dong et al., 2021; Zhou et al., 2021) retains the masked part, the new input can be formulated as $\hat{\mathbf{X}} = \mathbf{X} \odot \mathbf{S} + [\mathbf{M}] \odot (1 - \mathbf{S})$, where $\odot$ denotes the Hadamard product.

---

[1]Here we directly mask on the input RGB images, which is different with previous methods (Xie et al., 2022; Bao et al., 2021; Huang et al., 2022) which mask on the image patches (tokens).

| config | ViT-B (Dosovitskiy et al., 2020), Swin-B (Liu et al., 2021), Swin-L (Liu et al., 2021) |
|---|---|
| optimizer | AdamW (Loshchilov & Hutter, 2017) |
| base learning rate | 1.5e-4 |
| weight decay | 0.05 |
| optimizer momentum | $\beta_1, \beta_2 {=} 0.9, 0.95$ (Chen et al., 2020a) |
| batch size | 2048 |
| learning rate schedule | cosine (Loshchilov & Hutter, 2016), cosine (Loshchilov & Hutter, 2016), step (Xie et al., 2022) |
| warmup epochs | 10 |
| pre-training epochs | 800, 400, 400 |
| augmentation | RandomResizedCrop |

Table 11: Hyperparameters for pre-training ViT-B, Swin-B, and Swin-L on ImageNet-1K.

**Encoder Architecture.** We consider two typical transformers as the encoder (backbone) for pre-training, *i.e.*, ViT (Dosovitskiy et al., 2020) and Swin (Liu et al., 2021), which are both transferable to various downstream vision tasks. Therefore, the result can be directly compared with others in terms of the architecture.

**Decoder Architecture.** The latent feature extracted by encoder is then fed into the decoder, *i.e.*, a linear layer (Xie et al., 2022) or several transformer blocks (He et al., 2021), to predict the original pixels (He et al., 2021; Xie et al., 2022) or other targets (Wei et al., 2022; Bao et al., 2021; Dong et al., 2021) in the masked area.

**Prediction Target.** The targets can be the raw pixel values (He et al., 2021), Histograms of Oriented Gradients (HOG) (Wei et al., 2022), context encoded via dVAE (Bao et al., 2021; Chen et al., 2022b), *etc*.

## A.2 ABLATION STUDY ON MIM

**Mask strategy.** We analyze two typical mask strategies of the encoder, *i.e.*, MAE (He et al., 2021) and MIM (Xie et al., 2022). The former only operates on visible patches without [MASK] tokens, while the latter operates on the entire image patches. As shown in the top two rows of Table 10a, MIM achieves slightly better transfer performance compared to MAE, but operating on whole patches results in a heavier computational burden (as demonstrated in Figure 2 of our main paper). Moreover, the third row illustrates that masking on image patches (*e.g.*, $14{\times}14{\times}768$ in ViT-B (Dosovitskiy et al., 2020)) has almost the same effect as masking on the original image (*e.g.*, $224{\times}224{\times}3$). We further examine the impact of the [MASK]. We study two kinds of mask token, one with a learnable vector and the other set to zeros. We find that filling mask tokens with zeros degrades MAE performance by 0.4%, but has little impact on MIM. One primary reason is that the encoder in MIM can process the [MASK] earlier and more comprehensive than in MAE. In general, excluding the masked regions will not affect the final fine-tuning (transfer) result, except for the pre-training computational cost. **Mask size and mask ratio.** We study how different mask sizes and ratios affect the effectiveness of MIM in Table 10b. We observe that both isotropic and hierarchical architectures achieve their best results only when the mask size is equivalent to the patch size of the last stage of encoder. Notably, when the mask size is smaller than the patch size, MIM can still obtain comparable results, demonstrating its ability for representation learning. With an appropriate mask size, MIM remains stable with ratios varying from 0.5 to 0.85.

## A.3 IMPLEMENTATION DETAILS

### A.3.1 REVISITING OF MIM

In order to make MIM framework compatible with different vision backbones, we directly mask the original images in a block-wise manner (mask size can be adjusted in a large range), and retain all

| config | ViT-B (Dosovitskiy et al., 2020), Swin-B (Liu et al., 2021), Swin-L (Liu et al., 2021) |
|---|---|
| optimizer | AdamW (Loshchilov & Hutter, 2017) |
| base learning rate | 1.0e-3 |
| weight decay | 0.05 |
| optimizer momentum | $\beta_1, \beta_2$=0.9, 0.999 (Chen et al., 2020a) |
| layer-wise lr decay (Clark et al., 2020; Bao et al., 2021) | 0.7, 0.8, 0.75 |
| batch size | 1024 |
| learning rate schedule | cosine (Loshchilov & Hutter, 2016) |
| warmup epochs | 5 |
| training epochs | 100 |
| augmentation | RandAug (9, 0.5) (Cubuk et al., 2020) |
| label smoothing (Szegedy et al., 2016) | 0.1 |
| mixup (Zhang et al., 2017) | 0.8 |
| cutmix (Yun et al., 2019) | 1.0 |
| drop path rate (Huang et al., 2016) | 0.1, 0.1, 0.3 |

Table 12: Hyperparameters for fine-tuning ViT-B, Swin-B, and Swin-L on ImageNet-1K.

pixels during the pre-training stage. In this way, the encoder (*e.g.*, ViT in MAE (He et al., 2021) and Swin in SimMIM (Xie et al., 2022)) can be replaced by any architectures because the input image is of the same size as in supervised training. Here we study how resolution/target/encoder depth influence the MIM. All models are evaluated on two benchmarks, *i.e.*, ImageNet-1K (Deng et al., 2009) and COCO (Lin et al., 2014), which are commonly used in previous works (He et al., 2021; Bao et al., 2021; Xie et al., 2022; Huang et al., 2022; Dong et al., 2021; Chen et al., 2022b).

### A.3.2 IMAGENET EXPERIMENTS

Following common practice (He et al., 2021; Xie et al., 2022; Bao et al., 2021), we first conduct self-supervised pre-training on ImageNet-1K (Deng et al., 2009) training set without label, and then validate the proposed *FastMIM* by conducting end-to-end fine-tuning on downstream tasks including classification, object detection, instance segmentation, and semantic segmentation. All experiments are conducted on 8 V100 GPUs with PyTorch (Paszke et al., 2019).

**ViT architecture.** We follow the standard ViT architecture (Dosovitskiy et al., 2020). The encoder ends with an extra Layer Normalization (LN) (Ba et al., 2016). To match the different widths between encoder and decoder, we adopt a linear projection layer after the encoder following (He et al., 2021). Our *FastMIM* only adds absolute positional embeddings (the sine-cosine version) to the encoder inputs. And we retain the class token (Dosovitskiy et al., 2020) during our pre-training stage.

**Swin architecture.** We follow the standard Swin-B architecture (Liu et al., 2021). When pre-training with input images of size 128×128, the window size is set to 4 accordingly. When fine-tuning with input images of size 224×224, the window size is set to 7. And we simply leverage the "bicubic" interpolation to remap "relative position table" (Liu et al., 2021) when pre-trained window size mismatches with fine-tuned window size. As for Swin-L, we set the window size to 14 during fine-tuning stage following (Xie et al., 2022; Huang et al., 2022). *FastMIM* pre-trains Swin-L with 128×128 inputs and the window size is set to 8 accordingly. *FastMIM-P* gradually increases the input resolution during pre-training stage. We first initialize the window size to 14, and then interpolate the corresponding "relative position table" for different input resolutions.

**Pre-training.** The default setting is shown in Table 11. We simply use random resized cropping for data augmentation. We follow the official codes of ViT (Dosovitskiy et al., 2020) and Swin (Liu et al., 2021) to initialize corresponding blocks. We set the base learning rate to 1.5e-4, and the effective learning rate is scaled linearly: $lr = base\_lr \times batch\_size \ / \ 256$.

**Fine-tuning on ImageNet-1K.** The default setting is shown in Table 12. We follow previous practice (Bao et al., 2021; He et al., 2021) and use a layer-wise learning rate decay strategy (Clark et al., 2020; Bao et al., 2021) for fine-tuning. We fine-tune each backbone for 100 epochs with strong data augmentation including label smoothing (Szegedy et al., 2016), mixup (Zhang et al., 2017), and cutmix (Yun et al., 2019) following MAE (He et al., 2021) and SimMIM (Xie et al., 2022). The drop path rates (Huang et al., 2016) are set to 0.1/0.1/0.3 for ViT-B/Swin-B/Swin-L, respectively. To be noticed, we report the best top-1 accuracy through the grid search on base learning rate and layer-wise learning rate decay, as discussed in Sec. A.3.3 .

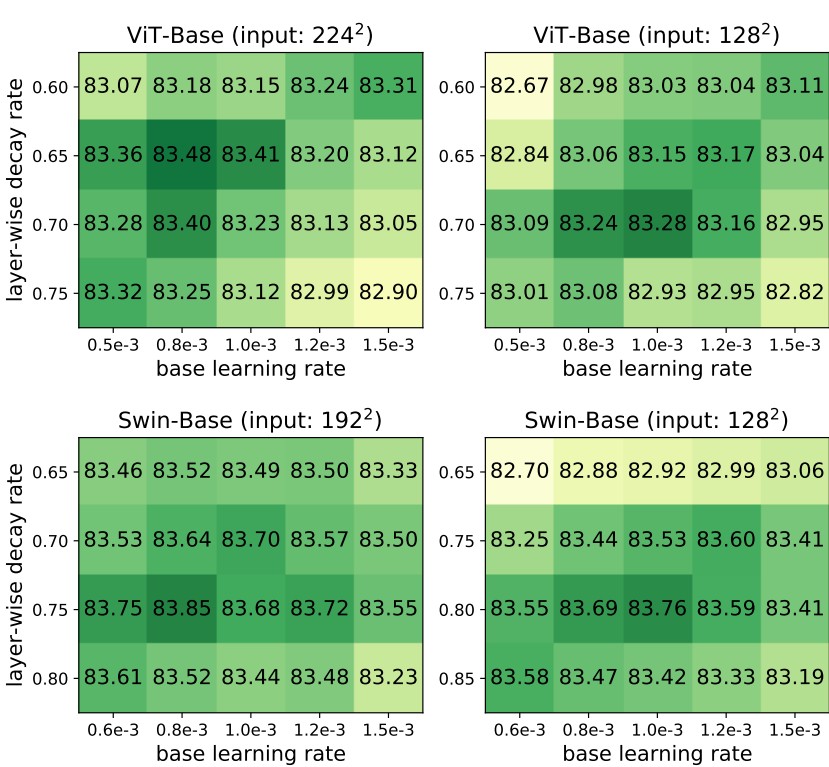

Figure 6: Grid search for fine-tuning hyperparameters. Top: ViT-B pre-trained with 400 epochs and pixel target. Bottom: Swin-B pre-trained with 400 epochs and pixel target. Deeper color indicates higher top-1 accuracy on ImageNet-1K validation set.

### A.3.3 Hyperparameters for Fine-tuning

To better adapt the pre-training formula to each model, we carefully sweep two hyperparameters via grid search in fine-tuning stage: (i) base learning rate (*blr*), and (ii) layer-wise decay rate (*ldr*), while keeping all others the same for all models. We conducted pilot experiments using ViT-B (Dosovitskiy et al., 2020) and Swin-B (Liu et al., 2021) pre-trained with our *FastMIM* to estimate reasonable hyperparameter ranges. We center a 3×3 grid at *blr*, *ldr* = {1.0e-3, 0.75} and use larger and smaller values around the center. If a local optimum is not found, *i.e.*, the best value is a boundary value, we expand the search. Figure 6 shows corresponding results of ViT-B and Swin-B.

### A.3.4 Object Detection and Segmentation on COCO

We adapt the ViT and Swin for the use of an FPN backbone (Lin et al., 2017) in Mask R-CNN (He et al., 2017). We follow three commonly used settings for fair comparison with other methods.

| config | ViT-B (Dosovitskiy et al., 2020), Swin-B (Liu et al., 2021), Swin-L (Liu et al., 2021) |
|---|---|
| optimizer | AdamW (Loshchilov & Hutter, 2017) |
| peak learning rate | 8e-5, 6e-5, 6e-5 |
| weight decay | 0.1, 0.05, 0.05 |
| optimizer momentum | $\beta_1, \beta_2$=0.9, 0.999 (Chen et al., 2020a) |
| batch size | 32 |
| learning rate schedule | cosine (Loshchilov & Hutter, 2016), step, step |
| warmup steps | 1500 |
| training epochs | 25 & 100, 36, 36 |
| input resolution | (1024, 1024) |
| drop path rate (Huang et al., 2016) | 0.1, 0.2, 0.3 |

Table 13: Hyperparameters for training ViT-B, Swin-B, and Swin-L on COCO benchmark.

| config | ViT-B (Dosovitskiy et al., 2020), Swin-B (Liu et al., 2021) |
|---|---|
| optimizer | AdamW (Loshchilov & Hutter, 2017) |
| peak learning rate | 1e-3, 3e-4 |
| weight decay | 0.05 |
| optimizer momentum | $\beta_1, \beta_2$=0.9, 0.999 (Chen et al., 2020a) |
| layer-wise lr decay (Clark et al., 2020; Bao et al., 2021) | 0.65, 0.9 |
| batch size | 16 |
| learning rate schedule | linear |
| warmup steps | 1500 |
| training steps | 160K |
| input resolution | (512, 512) |
| drop path rate (Huang et al., 2016) | 0.1 |

Table 14: Hyperparameters for training ViT-B and Swin-B on ADE20K benchmark.

| ep.\inp. | $96^2$ | $128^2$ | $160^2$ | $192^2$ | $224^2$ |
|---|---|---|---|---|---|
| 400 | 82.47 | 82.89 | 83.02 | 83.08 | 83.15 |
| 800 | 82.74 | 83.06 | 83.16 | 83.24 | 83.34 |

Table 15: Ablation on the input resolutions. Setting: MAE (He et al., 2021), ViT-B, raw pixel as prediction target. Top-1 Acc. is reported.

**MAE (He et al., 2021) setting.** We equally divide the 12 ViT-B blocks into 4 subsets and apply convolutions to upsample or downsample the intermediate feature maps for producing different scales following (Li et al., 2021; He et al., 2021). We train ViT-B with large-scale jitter (1024×1024 resolution, scale range [0.1, 2.0]) (Ghiasi et al., 2021), AdamW (Loshchilov & Hutter, 2017) with cosine learning rate decay, and drop path regularization for both 25 & 100 epochs, as shown in Table 13. More details can be found in (Li et al., 2021).

**GreenMIM (Huang et al., 2022) setting.** The learning rate setting is slightly different from Table 13. The peak learning rate is set to 1e-4 with a batch size of 16. The Swin-B is initialized with self-supervised pre-trained checkpoint via our *FastMIM*. More details can be found in (Huang et al., 2022).

**SimMIM (Xie et al., 2022) setting.** Table 13 shows the corresponding hyperparameters for Swin-B and Swin-L following (Xie et al., 2022). The window size for Swin-B is set to 7 and that for Swin-L

is 14. Notably, we choose upgraded Mask R-CNN (more details in Sec. 2.2 in (Li et al., 2021)) as basic framework and initialize the backbone with checkpoint fine-tuned on ImageNet-1K, following SimMIM (Xie et al., 2022). More details can be found in (Xie et al., 2022).

### A.3.5 SEMANTIC SEGMENTATION ON ADE20K

We use typical UperNet (Xiao et al., 2018) as the basic framework. We follow two previous settings to evaluate our *FastMIM*.

**MAE (He et al., 2021) setting.** We follow the semantic segmentation code of MAE (He et al., 2021) and BEiT (Bao et al., 2021). We fine-tune end-to-end for 100 epochs with a batch size of 16. We turn on relative position bias only during transfer learning, initialized as zero. We fine-tune end-to-end for 160K iterations using AdamW (Loshchilov & Hutter, 2017) optimizer with the peak learning rate of 3e-4, weight decay of 0.05. The ViT-B model is trained with input resolution of $512 \times 512$, as shown in Table 14.

**SimMIM (Xie et al., 2022) setting.** We follow the setting of SimMIM (Xie et al., 2022): a weight decay of 0.05, a batch size of 32, a layer-wise learning rate decay rate of 0.9, and a peak learning rate of 3e-4. The Swin-B model is trained with input resolution of $512 \times 512$, as shown in Table 14. We initialized the backbone with checkpoint after supervised fine-tuning on ImageNet-1K. In inference, a multi-scale test using resolutions that are $[0.75, 0.875, 1.0, 1.125, 1.25] \times$ of $512 \times 2048$ is employed.

| enc.\dec. | 1b256d | 1b512d | 1b768d | 4b256d | 4b512d | 4b768d | 8b256d | 8b512d | 8b768d |
|-----------|--------|--------|--------|--------|--------|--------|--------|--------|--------|
| ViT-B | 82.5 | 82.4 | 82.2 | 82.4 | 82.2 | N/A | 82.1 | 82.0 | N/A |
| ViT-L | 82.7 | 82.9 | 82.9 | 82.8 | 83.1 | 83.2 | 83.3 | 83.5 | 83.4 |
| Swin-B | 83.5 | 83.5 | 83.3 | 83.6 | 83.5 | 83.3 | 83.4 | 83.2 | N/A |
| Swin-L | 84.3 | 84.2 | 84.1 | 84.3 | 84.3 | 84.2 | 84.2 | 84.1 | 84.0 |

Table 16: **Ablation study on decoder size**. Pre-trained by our *FastMIM* framework with 100 epochs, *i.e.*, input size of $128^2$, HOG as prediction target. "1b256d" indicates one decoder block with 256-d width. Top-1 Accuracy is reported.

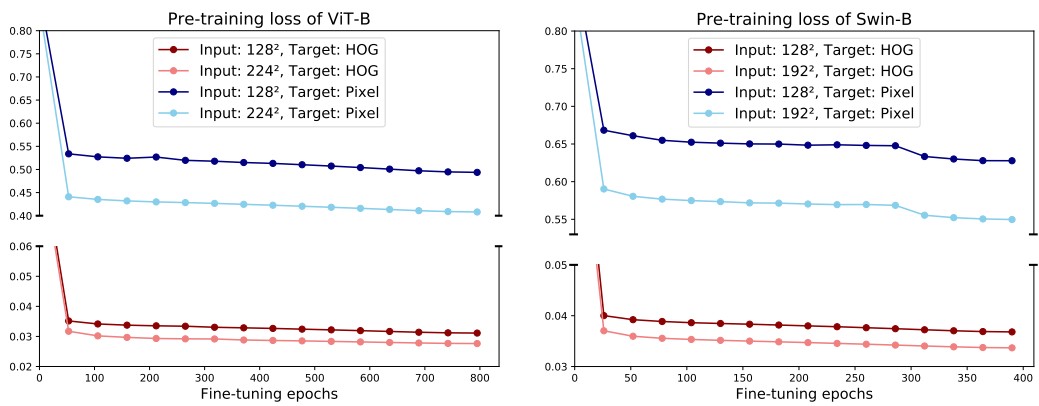

Figure 7: Pre-training loss on ImageNet-1K (Deng et al., 2009). ViT-B (left) is trained with 800 epochs and Swin-B (right) is trained with 400 epochs.

### A.4 MORE ABLATIONS ON BASIC COMPONENTS

**Reduce Input Resolution in MAE** Table 15 ablates how the pre-training epoch and input resolution impact the fine-tuning result of MAE framework (He et al., 2021). The final performance decreases when the input resolution is reduced. However, the performance drop resulted from decreasing input resolution of MAE from $224^2$ to $128^2$ is slightly larger when compared with MIM (Xie et al., 2022). We conjecture one main reason is that the MAE discards up to 75% patches during

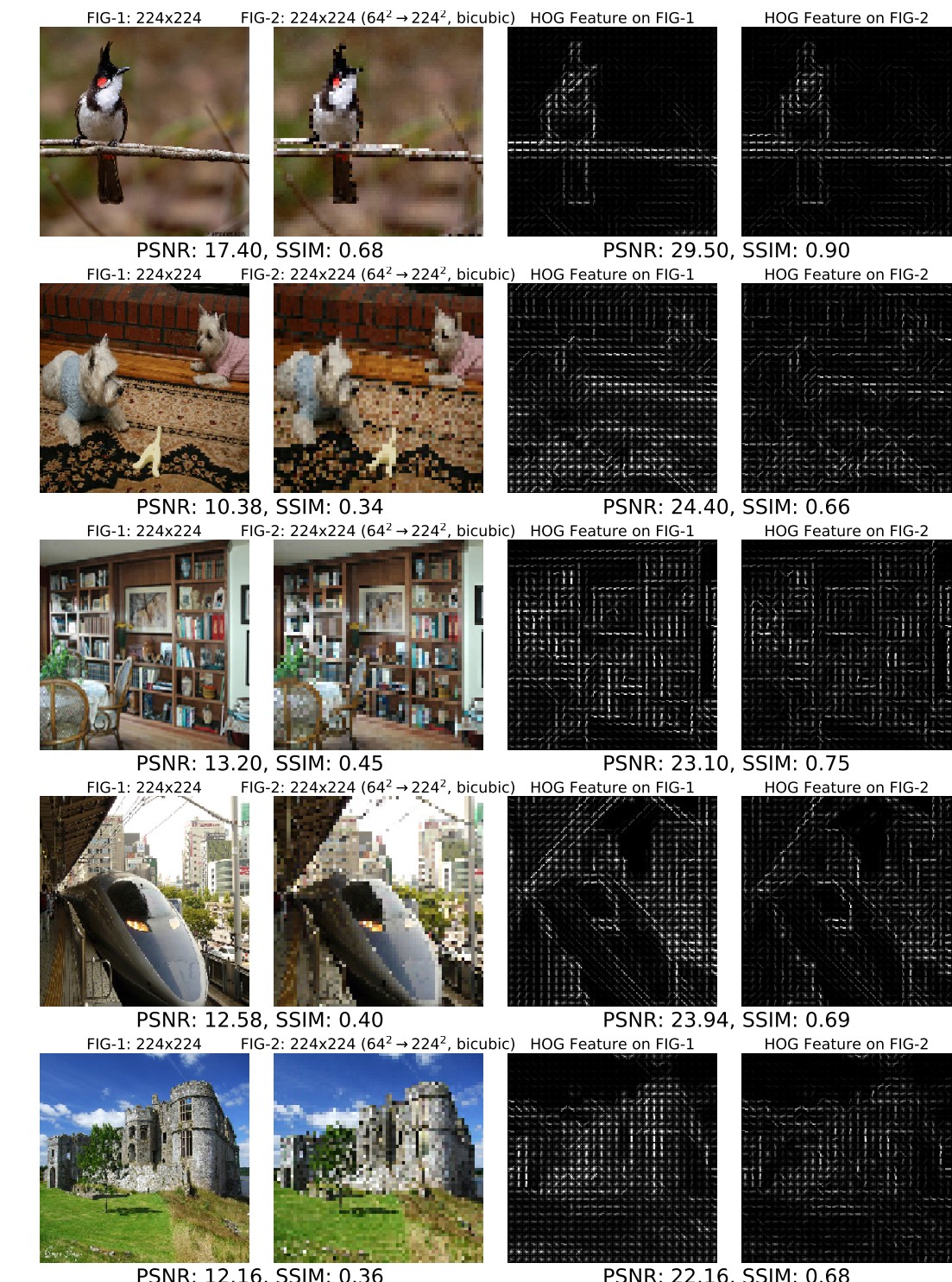

Figure 8: Visualization on pixel target and HOG target. Images are randomly chosen from ImageNet-1K (Deng et al., 2009). We choose PSNR(dB) and SSIM (Wang et al., 2004) to evaluate the similarity between two images (features). HOG target can preserve better texture information under low resolution input compared to pixel target.

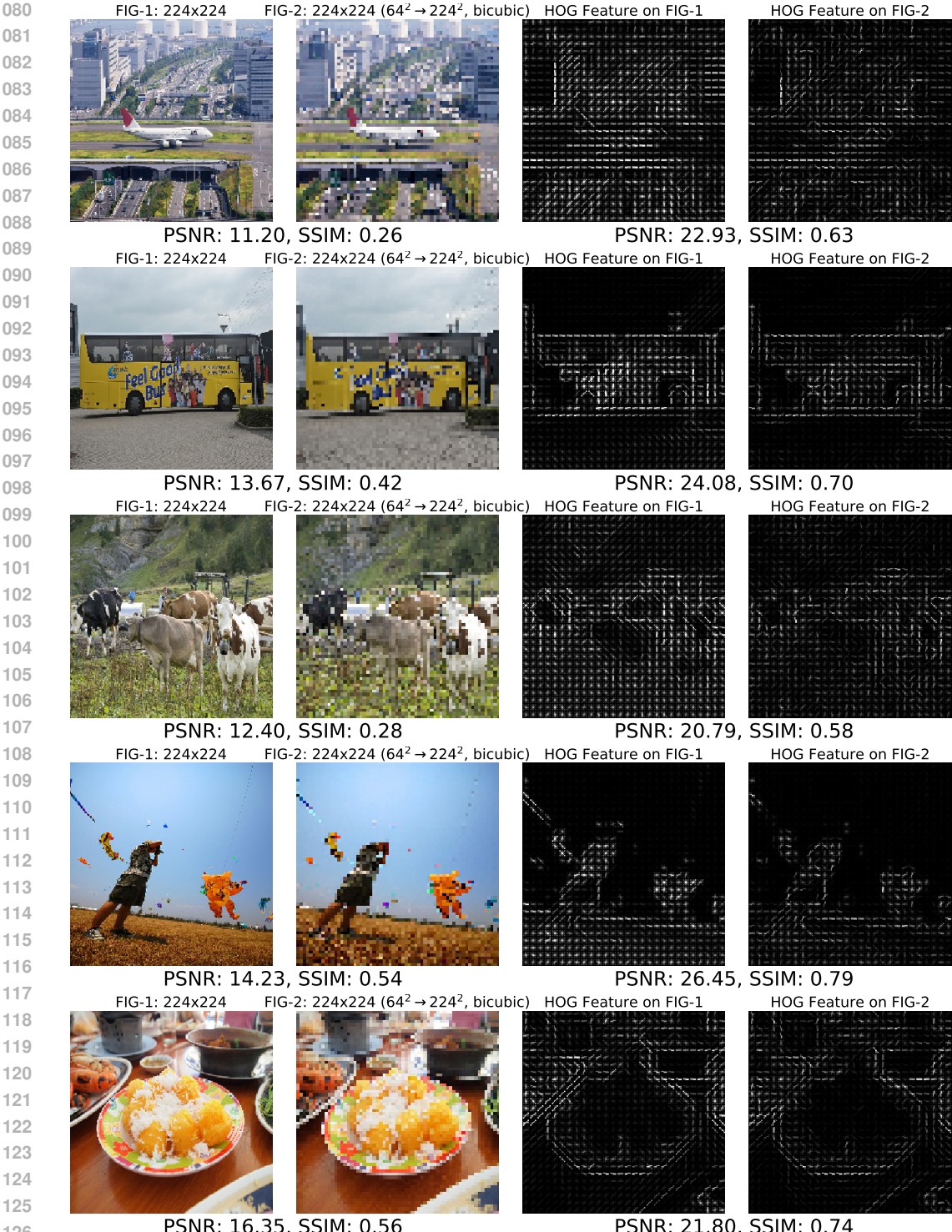

Figure 9: Visualization on pixel target and HOG target. Images are randomly chosen from COCO (Lin et al., 2014). We choose PSNR(dB) and SSIM (Wang et al., 2004) to evaluate the similarity between two images (features). HOG target can preserve better texture information under low resolution input compared to pixel target.

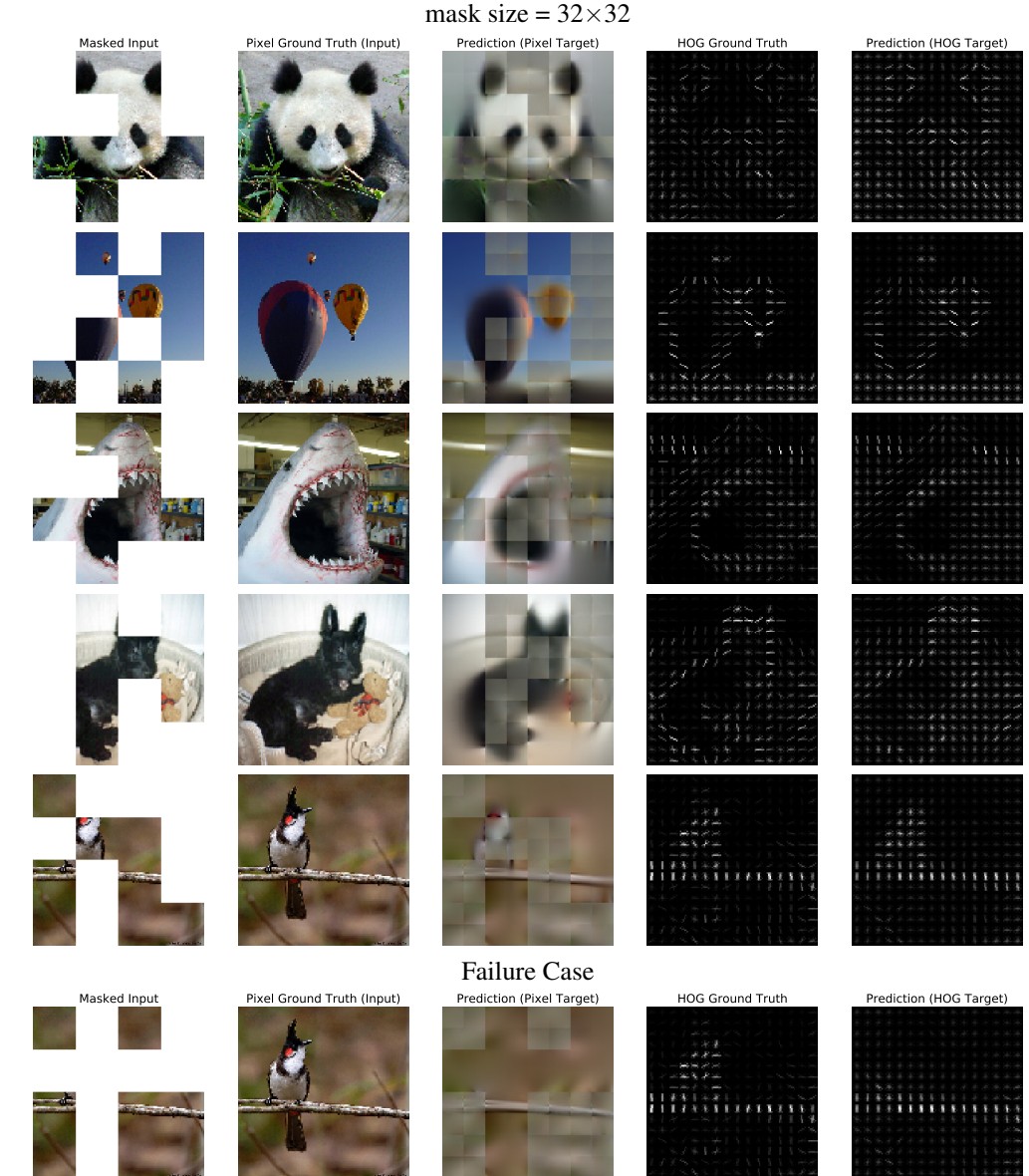

Figure 10: Pixel vs. HOG predictions (without normalization) on ImageNet-1K (Deng et al., 2009) validation set. Using an MIM trained on ImageNet. For each sample, we show the masked image, original input, prediction trained by pixel target, HOG ground truth, and prediction trained by HOG target. The unmasked regions are not used for loss and thus qualitatively poor.

pre-training stage, and reducing the input resolution will drastically decrease the number of visible patches, together with crucial position information for encoder. Although there is an extra absolute positional embedding added to the encoder input, the ability to capture (perceive) location information of MAE is inferior to MIM which retains the whole input patches.

**Ablation study on decoder size.** Table 16 ablates the effect of varying decoder designs. Intriguingly, the results suggest that different architectures prefer different settings and have opposite trends. ViT (Dosovitskiy et al., 2020) prefers a simple decoder for the base size and a complicated one for the large size. While Swin (Liu et al., 2021) seems to be robust with various decoder sizes and favors a simple one, which conforms with the observation in (Xie et al., 2022; Huang et al., 2022). In conclusion, the size of decoder should be properly aligned with the specific encoder.

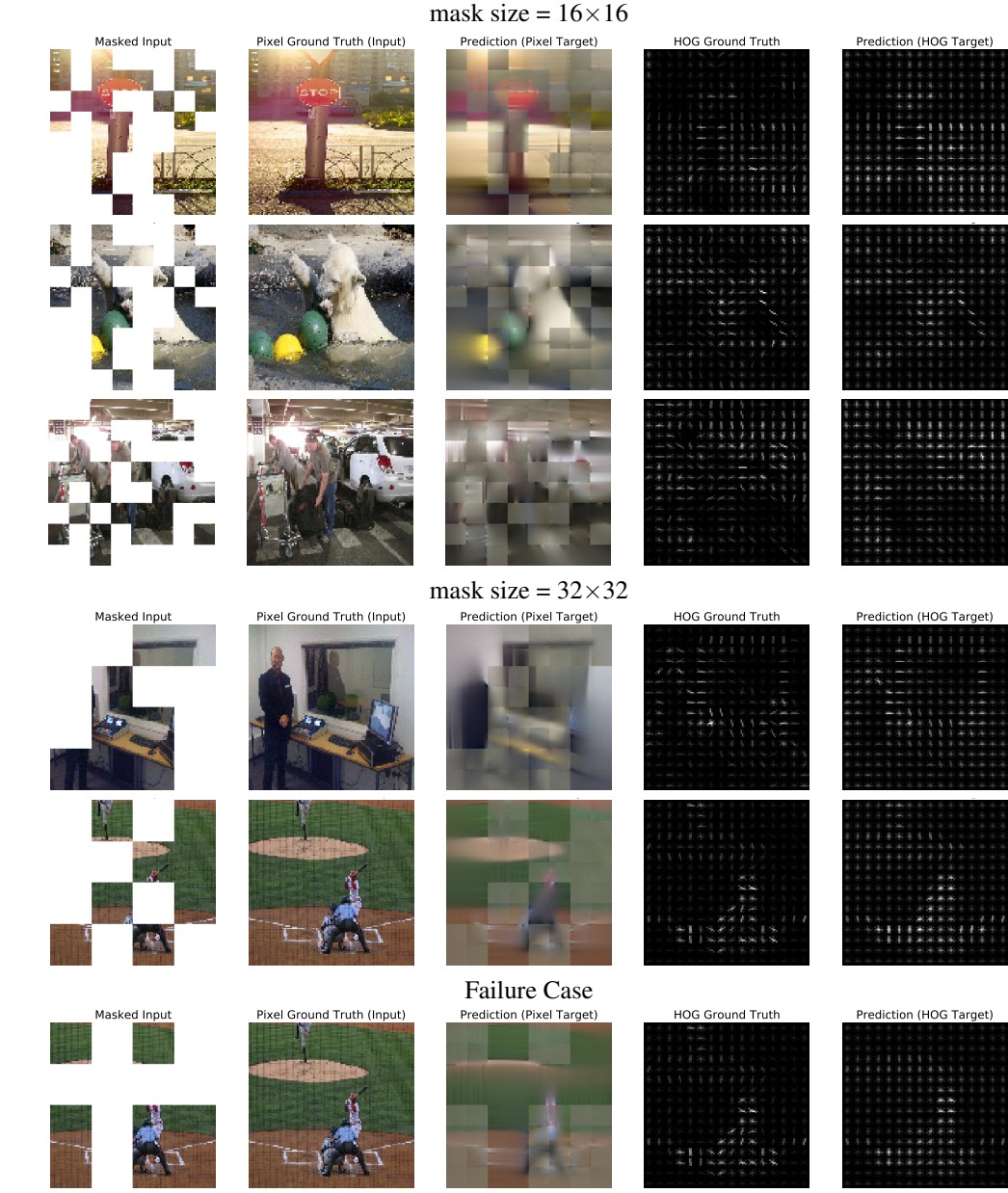

Figure 11: Pixel vs. HOG predictions (without normalization) on COCO (Deng et al., 2009) validation set. Using an MIM trained on ImageNet. For each sample, we show the masked image, original input, prediction trained by pixel target, HOG ground truth, and prediction trained by HOG target. The unmasked regions are not used for loss and thus qualitatively poor.

## A.5 MORE ABLATIONS ON THE HOG TARGET

**Pre-raining loss of the HOG target.** Figure 7 shows the pre-training losses of different input resolutions. We can find that HOG target can reduce the gap of loss values between different input resolutions. Besides, the absolute loss values of using HOG target are far smaller than those of using pixel target, demonstrating that HOG can effectively reduce the risk of ambiguity during reconstruction in MIM.

**Visualization of ground truth target.** Figure 8 and Figure 9 show more visualization results of pixel and HOG target on ImageNet-1K and COCO, respectively. Although reducing the image resolution can significantly expedite the training process, the crucial information, *e.g.*, detailed textures and edges, will be discarded when using pixel target. However, HOG is more invariant to the resolution changes, which is suitable for our *FastMIM*.

**Visualization of predicted targets.** We qualitatively compare the reconstruction result of pixel target with HOG target as shown in Figure 10 and Figure 11. We can find that both pixel and HOG predictions are semantically plausible to some extent. However, pixel targets suffer from large errors caused by ambiguous problems (Wei et al., 2022), while HOG is more robust to ambiguity. As shown in the second row in Figure 10, the model trained via pixel target predicts the balloon as dark blue, which is in fact red in the top area. This wrong prediction can result in a high loss penalty, which can also increase the difficulty of training. This is also affirmed by MaskFeat (Wei et al., 2022) and is also the main reason for MaskFeat to leverage HOG feature as the prediction target. In addition to above reason, we demonstrate that HOG is more invariant to the resolution changes when compared with pixel target. Therefore, HOG target is naturally more suitable for our *FastMIM*.

