# OpenReview forum: "Empirical Study on Enhancing Efficiency in Masked Image Modeling Pre-training"
_ICLR.cc/2025/Conference — Submitted to ICLR 2025_

### Official Review · Reviewer_kS5H · 2024-10-31

**Soundness:** 3
**Presentation:** 3
**Contribution:** 2
**Rating:** 5
**Confidence:** 4

**Summary:**

This work conducts an empirical study on enhancing training efficiency in masked image modeling (MIM). The authors introduce a simple and versatile framework that expedites MIM by using low-resolution images and reconstructing HOG features.

**Strengths:**

1. Improving the efficiency of MIM pre-training is a good topic and has significant practical value.
2. This paper contains numerous experiments and reveals some interesting findings. For instance, the authors find that discarding the last few layers can speed up the training process without affecting fine-tuning performance.

**Weaknesses:**

1. My main concern lies in the lack of novelty. The problem of low efficiency in MIM has been pointed out by previous works as the paper stated. The idea of using low-resolution images to expedite training is too simple and has been implemented in previous works. Additionally, the superiority of reconstructing HOG features has been demonstrated by its original paper.

2. The proposed method has small gap for the performance of models with high capacity, e.g., 0.3% for Swin-L. It significantly diminishes the advantages of the proposed approach as one may just reduce training epochs to achieve this trade-off. For smaller models, it is more efficient to reconstruct features of pre-trained larger models like a kind of knowledge distillation.

3. The proposed FastMIM-P progressively increases resolution to alleviate the above problem. However, the technical novelty of this approach is still limited. And the resolution and training schedule need to be carefully designed to achieve a better space-time trade-off as the authors stated.

4. In Table 4, it can be seen that the authors try to demonstrate the generalization across different visual backbones. Nevertheless, the performance gain on some backbones lags behind recent MIM works due to the lack of integration of advanced technology such as masked convolutions.

**Questions:**

In Table 4, it is more meaningful to compare total training time, including the pre-training time and fine-tuning time.

---

> ### Author Response · Authors · 2024-11-23
> **Response to Reviewer kS5H (part1/2)**
>
> **W1: The problem of low efficiency in MIM has been pointed out by previous works as the paper stated. The idea of using low-resolution images to expedite training is too simple and has been implemented in previous works. Additionally, the superiority of reconstructing HOG features has been demonstrated by its original paper.**
>
> A1: Thank you for your valuable feedback. We respectfully highlight the following key aspects to address these concerns and clarify the contributions of our paper:
>
> 1. **Positioning of our paper as a Practical Paradigm**
>
>    - **First**, weFastMIM is positioned as an **empirical study**, inspired by works [1,2], that thoroughly examines prior MIM methods and identifies a critical research gap: **self-supervised pre-training demands enormous computational resources**, making it impractical for single deep learning machines. Our philosophy is to break the barrier of costly pre-training by providing a solution that is both **efficient** and **deployable**, enabling broader accessibility to MIM research.
>
>    [1] Xinlei Chen, Saining Xie, et al. "An empirical study of training self-supervised vision transformers." ICCV 2021.
>
>    [2] Zi-Yi Dou, Yichong Xu, et al. "An empirical study of training end-to-end vision-and-language transformers." CVPR 2022.
>
> 2. **Key Contributions**
>    While both small input sizes and HOG targets have been explored independently in prior works, our contributions are two-fold:
>
>    - **First**, we demonstrate for the first time that combining these two factors leads to a **qualitative transformation in efficiency and practicality** for MIM. Unlike previous works that consider these aspects in isolation, our study reveals the synergistic effect of small input sizes and HOG targets in addressing computational bottlenecks, offering a clear and actionable pathway for efficient pre-training.
>
>    - **Second**, our motivations are fundamentally different from related works such as SimMIM. For instance:
>
>      - **SimMIM** only **slightly reduces** input size (from $224^2$ to $192^2$), aiming to mitigate memory overhead. This minor reduction, however, is insufficient to drastically improve training efficiency.
>
>      - In contrast, **FastMIM** adopts a **significantly smaller input size**, reducing the resolution by a factor of four. This results in a **substantial acceleration** in training and memory savings, as shown in Table below:
>
>        | Case    | GPU  | Memory | # GPUs | PT Days |
>        | ------- | ---- | ------ | ------ | ------- |
>        | EVA     | A100 | ~27GB  | 128    | ~14.5   |
>        | SwinV2  | A100 | <40GB  | 8      | ~20     |
>        | CAE     | V100 | <32GB  | 8      | ~12     |
>        | SimMIM  | V100 | ~28GB  | 8      | ~7      |
>        | FastMIM | V100 | ~14GB  | 8      | ~1.6    |
>
>        FastMIM reduces memory consumption to **14GB** on a V100 GPU (compared to $\sim$28GB for SimMIM) and achieves pre-training in **1.6 days**, compared to $\sim$7 days for SimMIM, $\sim$12 days for CAE, and $\sim$14.5 days for EVA. These results clearly demonstrate the practicality and scalability of our approach.
>
> 3. To substantiate our claims, we conducted extensive experiments:
>
>    - **Robustness to Input Resolution**: Our findings (Table 1a in the main paper) indicate that a wide range of input resolutions yield comparable performance. This highlights that the efficacy of MIM is preserved even with smaller inputs, a result not emphasized in prior works.
>
>    - **Significance of Input Size in Pre-training**: We explicitly analyze the impact of input resolution during pre-training and fine-tuning (Table 8 in the main paper), providing new insights into why reducing resolution during pre-training improves both efficiency and downstream performance.

---

> > ### Author Response · Authors · 2024-11-24
> > **Response to Reviewer kS5H (part2/2)**
> >
> > **W2: It significantly diminishes the advantages of the proposed approach as one may just reduce training epochs to achieve this trade-off. For smaller models, it is more efficient to reconstruct features of pre-trained larger models like a kind of knowledge distillation.**
> >
> > A2: Thanks for your comments.
> >
> > - **On reducing training epochs:**
> >    While reducing training epochs can indeed shorten the pre-training time, it does not address the memory overhead issue. Large memory consumption remains a bottleneck for many researchers and practitioners working with limited resources, as the memory demand is largely independent of the number of epochs. Additionally, reducing training epochs often results in a more significant performance degradation compared to our approach of reducing input resolution.
> >
> > - **On reconstructing features of pre-trained larger models:**
> >    Thanks for your comments. We agree that reconstructing features from pre-trained larger models, akin to knowledge distillation, can be an effective strategy for smaller models, this does not conflict with the approach we propose.. To address this, we have added a discussion with such method [1] in the updated manuscript (Sec. 5.6) and conducted experiments to evaluate this approach. We show that our method is complementary to such distillation based reconstruction techniques.
> >
> > [1] Bai, Yutong, et al. "Masked autoencoders enable efficient knowledge distillers." CVPR  2023.
> >
> > **W3: The technical novelty of this approach is still limited. And the resolution and training schedule need to be carefully designed to achieve a better space-time trade-off as the authors stated.**
> >
> > A3: In this paper, we aim to emphasize that input resolution during the pretraining stage is not as critical as commonly assumed. While we do not propose particularly novel or fancy methods, our focus is on providing practical baselines and valuable insights. We believe the significance of our empirical study lies in its comprehensive evaluation and its practical contributions to the field.
> >
> > **W4: the performance gain on some backbones lags behind recent MIM works due to the lack of integration of advanced technology such as masked convolutions.**
> >
> > A4: We have added a comparison of our method with the results from Masked Convolution [1] and updated Table 4 in the revised version. By reducing the input resolution, we can effectively shorten the pretraining time.
> >
> > [1] Woo, Sanghyun, et al. "Convnext v2: Co-designing and scaling convnets with masked autoencoders." CVPR 2023.
> >
> > **Q4: In Table 4, it is more meaningful to compare total training time, including the pre-training time and fine-tuning time.**
> >
> > A4: Thanks for your suggestion. We have updated the Table 4 accordingly.

---

### Official Review · Reviewer_teGB · 2024-11-01

**Soundness:** 3
**Presentation:** 3
**Contribution:** 1
**Rating:** 3
**Confidence:** 5

**Summary:**

This paper introduces FastMIM, a simple and versatile framework that accelerates masked image modeling pre-training through two steps: (i) pre-training vision backbones using low-resolution input images, and (ii) reconstructing Histograms of Oriented Gradients (HOG) features instead of original RGB values. FastMIM-P, a variant, progressively increases input resolution during pre-training. Additionally, it emphasizes the importance of shallow layers during pre-training and suggests discarding the last few layers to speed up training without affecting fine-tuning performance. FastMIM enables efficient pre-training of any vision backbone, and has made lots of verification experiments.

**Strengths:**

The paper provides valuable insights into the design of MIM frameworks, including the importance of input resolution, the role of shallow layers during pre-training, and the stability of HOG features when transferring resolution.

**Weaknesses:**

1. The main contributions of the proposed method, i.e., the HOG reconstruction target and low-resolution input, have been extensively discussed in previous papers such as MaskFeat and SimMIM. This makes the proposed method appear less novel.
2. The empirical study mainly focuses on input resolution, training epochs, prediction targets, and the number of decoder/encoder layers. However, these aspects are common knowledge in MIM research, thus providing limited contribution to the community.
3. When verifying the importance of shallow layers during pre-training, the linear probing accuracy is missing, making the argument less convincing.

**Questions:**

See weaknesses part.

---

> ### Author Response · Authors · 2024-11-23
> **Response to Reviewer teGB**
>
> **W1: The proposed method appear less novel.**
>
> A1: Thank you for your valuable feedback. We respectfully highlight the following key aspects to address these concerns and clarify the contributions of our paper. Our paper is positioned as an **empirical study**, inspired by works [1,2], that thoroughly examines prior MIM methods and identifies a critical research gap: **self-supervised pre-training demands enormous computational resources**, making it impractical for single deep learning machines. Our philosophy is to break the barrier of costly pre-training by providing a solution that is both **efficient** and **deployable**, enabling broader accessibility to MIM research.
>
> [1] Xinlei Chen, Saining Xie, et al. "An empirical study of training self-supervised vision transformers." ICCV 2021.
>
> [2] Zi-Yi Dou, Yichong Xu, et al. "An empirical study of training end-to-end vision-and-language transformers." CVPR 2022.
>
> The two key factors behind our method, **(i) smaller input size** and **(ii) HOG target**, have been proposed in earlier works. However, our contributions lie in two critical differences:
>
> - **Combination of Factors:** We are the first to highlight that the combination of these two factors enables a significantly more **efficient and practical** MIM framework, a direction not yet explored in the literature.
> - **Different Motivations:** The motivations of our *FastMIM* differ fundamentally from those of SimMIM and MaskFeat. While SimMIM **slightly reduces** input size to mitigate memory overhead, our *FastMIM* deliberately adopts a much smaller input resolution to expedite training while maintaining comparable performance. Additionally, our experiments reveal that a wide range of input resolutions achieve similar fine-tuning results, underscoring the flexibility of our approach.
>
> ### Comparison with SimMIM
>
> - **Motivation:** SimMIM focuses on improving representation learning performance by analyzing various components. In contrast, *FastMIM* is designed to provide a **faster training paradigm** with reduced memory consumption.
> - **Input Resolution:** SimMIM pre-trains ViT-B with a 224$^2$ input size and Swin-B/L/H with 192$^2$ input size to address computational overhead but does not explore the effects of further reducing input resolution. By directly lowering the input resolution to $128^2$, *FastMIM* significantly reduces the number of patches, leading to faster training with minimal performance loss.
> - **Target Resolution:** Section 4.1.4 of SimMIM ablates the prediction (output) resolutions rather than input resolution, which offers negligible computational savings. The input resolution in SimMIM remains **fixed** at 192$^2$, limiting its potential for accelerating training.
>
> ### Comparison with MaskFeat
>
> Our primary goal is to accelerate the pre-training process. Using low-resolution inputs directly, however, often causes significant performance drops when employing pixel targets commonly used in MIM.
>
> Inspired by MaskFeat, which demonstrates the robustness of HOG targets against color ambiguities, we further elaborate on **HOG's invariance to resolution changes**. Our experiments and visualizations reveal that HOG retains better fine-tuning accuracy and achieves lower pre-training loss under low-resolution inputs compared to pixel targets. Thus, we adopt HOG to mitigate the loss of critical information (e.g., textures and edges) that typically occurs with smaller input resolutions, achieving both **efficiency** and **robustness** in the pre-training process.
>
> **W2: these aspects are common knowledge in MIM research, thus providing limited contribution to the community.**
>
> A2: Our primary focus is distinct: we aim to demonstrate that significantly reducing the input resolution during pre-training has minimal impact on final fine-tuning performance.
>
> This insight is both practical because it enables the development of **faster and more memory-efficient baselines** without compromising accuracy. By systematically evaluating the interplay of input resolution and prediction targets, we provide a streamlined approach for pre-training, which can serve as a solid reference for future work.
>
> While our contributions may not introduce highly novel methods, we believe the value of this work lies in its **practical guidance** and the ability to deliver competitive baselines that are both efficient and effective.
>
> **W3: When erifying the shallow layers during pre-training, the linear probing accuracy is missing.**
>
> A3: Since we used fewer layers during pretraining, we introduced new layers (random initilized) during fine-tuning. This demonstrates that the layers closer to the end do not necessarily require self-supervised pretraining, rather than suggesting that our network inherently does not need the final layers. Our experiments have validated this hypothesis. However, because the final layers were not pretrained, it is not feasible to perform corresponding experiments using linear probing.

---

### Official Review · Reviewer_6cgN · 2024-11-03

**Soundness:** 3
**Presentation:** 3
**Contribution:** 3
**Rating:** 6
**Confidence:** 5

**Summary:**

Masked Image Modeling (MIM) is a type computaional intentive pre-training methods. To  reduce the computational overhead of MIM, this paper proposed FastMIM to speeds up pre-training through two main strategies: (i) using low-resolution input images and (ii) reconstructing Histograms of Oriented Gradients (HOG) features instead of the original RGB values. FastMIM is compatible with both hierarchical and non-hierarchical transformer models. Experiments on ImageNet-1k, MS COCO and ADE20K are conducted.

**Strengths:**

- The visual analysis presented in Figure 3 is clear and compelling.

-  The paper demonstrates a strong motivation for the work, introducing efficient strategies to mitigate the high computational cost associated with MIM pre-training.

-  The proposed method shows promising results on the MS COCO object detection task, highlighting its potential effectiveness.

**Weaknesses:**

- From Figure 4, the accuracy does not demonstrate a clear saturation trend. Providing the fine-tuning accuracy after 1600 pre-training epochs would make the results more convincing.

- Missing citation: In lines 190–200 (Encoder depth in pre-training), the observation that discarding the last several layers (blocks) in pre-training yields nearly the same performance has been previously noted in MIRL [1].

- While the reviewer acknowledges the efficiency of the proposed method, the performance gains of FastMIM on ImageNet-1K and ADE20K are marginal.

- Experimental results for ViT-L are not provided.

[1] Huang, Guoxi, Hongtao Fu, and Adrian G. Bors. "Masked image residual learning for scaling deeper vision transformers." Advances in Neural Information Processing Systems 36 (2023): 57570-57582.

**Questions:**

- Please refer to the Weaknesses.

- Additionally, the reviewer is concerned about the effectiveness of FastMIM for large-scale models such as ViT-L and Swin-L, as these models tend to be more data-hungry. Reducing the input size decreases the number of visual tokens, which in turn significantly reduces the totall number of possible mask patterns. This is equivalent to a reduction in input diversity.

If the authors can address the reviewer's questions one by one, the reviewer is willing to consider raising the score.

---

> ### Author Response · Authors · 2024-11-23
> **Response to Reviewer 6cgN**
>
> **W1 : From Figure 4, the accuracy does not demonstrate a clear saturation trend. Providing the fine-tuning accuracy after 1600 pre-training epochs would make the results more convincing.**
>
> A1: Thanks for your advice. We have updated the results based on ViT-B and will provide the Swin-B results once the experiments are completed.
>
> **W2: Missing citation.**
>
> A2: Thank you for the reminder. We have updated the revised paper accordingly.
>
> **W3: While the reviewer acknowledges the efficiency of the proposed method, the performance gains of FastMIM on ImageNet-1K and ADE20K are marginal.**
>
> A3: Our main contribution is to expedite and alleviate the computational overhead for practical MIM-based pretraining. As shown in Figure 1 and Figure 2, FastMIM reduces memory consumption by approximately 3x and accelerates the pretraining procedure by around 5x, while achieving comparable or better downstream performance compared to previous frameworks.
>
> **W4 & Q1: The effectiveness of FastMIM for large-scale models such as ViT-L and Swin-L**
>
> A4: Thank you for your valuable suggestion. In the revised version, we have included experimental results for ViT-L in Table-3. Consistent with the trend observed from Swin-B to Swin-L, we find that reducing the input resolution results in slightly more pronounced accuracy degradation as the model scales up from ViT-B to ViT-L. This further validates that larger models indeed tend to be more data-hungry, and a smaller number of input tokens may lead to insufficient pretraining. However, by leveraging the progressively enlarged input resolution strategy, we effectively mitigate this issue and achieve significant improvements in accuracy for larger models. The updated discussion in the revised manuscript further elaborates on these findings.
>
> | Framework | Model | # Params | FT Ep. | Hours/Ep. | PT Hours | Top-1 |
> | --------- | ----- | -------- | ------ | --------- | -------- | ----- |
> | MAE       | ViT-L | 307M     | 1600   | 2.0       | 3260     | 85.9  |
> | FastMIM   | ViT-L | 307M     | 800    | 1.3       | 1062     | 85.1  |
> | FastMIM-P | ViT-L | 307M     | 800    | 1.8       | 1434     | 85.7  |
>
> ##

---

> > ### Comment · Reviewer_6cgN · 2024-11-26
> >
> > Thank you for your response.
> >
> > All my concerns are well-addressed.
> >
> > The idea is simple and efficient. I am looking forward to your source code.
> >
> > I raise my score to 6.
> >
> > Good luck!

---

> > > ### Author Response · Authors · 2024-11-26
> > >
> > > Thank you for taking the time to provide valuable suggestions on our paper! We truly appreciate your comments and support. We will release the code in the near future.
> > >
> > >
> > > Best regards,
> > >
> > > Authors

---

### Official Review · Reviewer_cXUf · 2024-11-05

**Soundness:** 3
**Presentation:** 4
**Contribution:** 3
**Rating:** 5
**Confidence:** 5

**Summary:**

The paper introduces FastMIM, a framework that speeds up masked image modeling (MIM) pre-training by using low-resolution images and reconstructing HOG features instead of RGB values. FastMIM-P further improves performance by progressively increasing input resolution during pre-training. This approach maintains good performance and accelerates training compared to previous methods.

**Strengths:**

1. Speedup in Pre-training: By using low-resolution input images, FastMIM significantly reduces pre-training time.
2. High Accuracy: Despite accelerating the training process, FastMIM maintains high accuracy.
3. The paper provides detailed experimental results, which help in understanding the robustness and reliability of the proposed method. The writing is clear and logically structured, making it easy to follow the methodology and results.

**Weaknesses:**

The contribution is limited. The paper improves MIM by using low-resolution images as input and HOG features as learning targets, with the advantages of HOG features already validated in the maskfeat work. Using low-resolution image input can accelerate the pre-training process without causing significant performance degradation, which indeed can serve as an acceleration trick for MIM pre-training. However, since model pre-training does not occur frequently in practical applications, the time cost of pre-training is acceptable. The main purpose of pre-training is to obtain better foundational model representations.

**Questions:**

1. The caption is incorrect in Figure 4, the text in the top left corner of the figure should annotate the line segment as HOG instead of pixel.
2. How does the method perform compared to other methods on larger pretraining datasets such as IM-21K or with longer pretraining epochs? If it can be demonstrated that Fast-MIM has better scaling-up capabilities, then its accelerated training would be more advantageous.

---

> ### Author Response · Authors · 2024-11-23
> **Response to Reviewer cXUf**
>
> **W1: However, since model pre-training does not occur frequently in practical applications, the time cost of pre-training is acceptable. The main purpose of pre-training is to obtain better foundational model representations.**
>
> A1: Thanks for your valuable comments. I agree with your point that, on a macro level, the time cost of pretraining is acceptable. However, we would like to emphasize that the motivation of this paper is to provide a baseline method that makes current pretraining more **efficient** and **deployable**. Our goal is to enable relevant research to be conducted on smaller, commercially available GPUs, rather than requiring high-end GPUs like the A100, while also significantly saving time.
>
> FastMIM is positioned as an empirical study, inspired by works [1,2], that thoroughly examines prior MIM methods and identifies a critical research gap: self-supervised pre-training demands enormous computational resources, making it impractical for single deep learning machines. Our philosophy is to break the barrier of costly pre-training by providing a solution that is both efficient and deployable, enabling broader accessibility to MIM research.
>
>    [1] Xinlei Chen, Saining Xie, et al. "An empirical study of training self-supervised vision transformers." ICCV 2021.
>
>    [2] Zi-Yi Dou, Yichong Xu, et al. "An empirical study of training end-to-end vision-and-language transformers." CVPR 2022.
>
> Our contributions are two-fold:
>
>    - We demonstrate for the first time that combining these two factors leads to a qualitative transformation in efficiency and practicality for MIM. Unlike previous works that consider these aspects in isolation, our study reveals the synergistic effect of small input sizes and HOG targets in addressing computational bottlenecks, offering a clear and actionable pathway for efficient pre-training.
>
>    - Our motivations are fundamentally different from related works such as SimMIM. For instance:
>
>      - SimMIM only slightly reduces input size (from $224^2$ to $192^2$), aiming to mitigate memory overhead. This minor reduction, however, is insufficient to drastically improve training efficiency.
>
>      - In contrast, FastMIM adopts a significantly smaller input size, reducing the resolution by a factor of four. This results in a substantial acceleration in training and memory savings, as shown in Table below:
>
>        | Case    | GPU  | Memory | # GPUs | PT Days |
>        | ------- | ---- | ------ | ------ | ------- |
>        | EVA     | A100 | ~27GB  | 128    | ~14.5   |
>        | SwinV2  | A100 | <40GB  | 8      | ~20     |
>        | CAE     | V100 | <32GB  | 8      | ~12     |
>        | SimMIM  | V100 | ~28GB  | 8      | ~7      |
>        | FastMIM | V100 | ~14GB  | 8      | ~1.6    |
>
>        FastMIM reduces memory consumption to 14GB on a V100 GPU (compared to $\sim$28GB for SimMIM) and achieves pre-training in 1.6 days, compared to $\sim$7 days for SimMIM, $\sim$12 days for CAE, and $\sim$14.5 days for EVA. These results clearly demonstrate the practicality and scalability of our approach.
>
> **Q1: The caption is incorrect in Figure 4, the text in the top left corner of the figure should annotate the line segment as HOG instead of pixel.**
>
> A1: Thanks for raising this point, we have corrected the corresponding errors in our revision.
>
> **Q2: How does the method perform compared to other methods on larger pretraining datasets such as IM-21K or with longer pretraining epochs?**
>
> A2: Thanks for your suggestion. In Table 8, we compare pretraining on ImageNet-21K and ImageNet-1K, and we observe that larger pretraining datasets can significantly improve performance. But using SimMIM on the 22K dataset with 192² input resolution requires around 30 days to complete pretraining, which demands substantial computational resources, and we were unable to obtain results within this time frame, we will update this result once the experiments are completed. Additionally, we have included results on ViT-L in Table 3, further demonstrating the effectiveness of our method on larger models.

---

> > ### Comment · Reviewer_cXUf · 2024-12-03
> >
> > Thank you for your detailed response. However, the conclusions drawn from the empirical study in this paper are not particularly innovative. Reducing the resolution of the input indeed results in performance degradation as the model scales up, making the conclusion an expected trade-off between performance and speed. This method also does not demonstrate any performance advantages in scaling up, aside from training speed. Additionally, the finding that HOG is a better target for masked image modeling than RGB values has been validated in previous works. Therefore, I will maintain my original score.

---

### Official Review · Reviewer_Yv3g · 2024-11-07

**Soundness:** 3
**Presentation:** 3
**Contribution:** 3
**Rating:** 6
**Confidence:** 4

**Summary:**

This paper introduces FastMIM, a framework that accelerates masked image modeling (MIM) by using low-resolution input images and Histograms of Oriented Gradients (HOG) features for pre-training, achieving improved efficiency and accuracy in transfer learning tasks. The authors demonstrate that FastMIM achieves superior top-1 accuracy on ImageNet-1K and speeds up training by approximately 5×, with additional insights on resolution variation, layer importance, and the stability of HOG features over RGB values.

**Strengths:**

1.	The paper identifies and leverages low-resolution input images to significantly reduce both the pre-training time and memory usage.
2.	By reconstructing Histograms of Oriented Gradients (HOG) features, the method preserves crucial texture information that is often lost with lower resolutions.
3.	The proposed FastMIM framework is validated through extensive experiments.

**Weaknesses:**

1.	Risk of Information Loss: By reducing the resolution and replacing RGB pixels with HOG features, some fine-grained details may be lost, potentially affecting performance in tasks that require precise image analysis.
2.	Task-Specific Adaptability: While HOG features perform stably in general vision tasks, they may not be suitable for tasks that require precise texture or color information, such as image generation or super-resolution reconstruction.
3.	Limitations of Method Generality: The performance of this method during pre-training depends on the stability of HOG features, and it may not be suitable for model architectures or specialized tasks that require a broader range of features.

**Questions:**

1.	The method is well-suited for a wide range of vision tasks, particularly in environments with limited computational resources. However, considering its general applicability and performance across all tasks, especially those requiring high-detail image analysis, further improvements may be necessary to minimize information loss. If the above limitations can be overcome, particularly the issues with task-specific adaptability, this work holds great potential.

---

> ### Author Response · Authors · 2024-11-23
> **Response to Reviewer Yv3g**
>
> **Q1: Risk of Information Loss**
>
> A1: Thank you for raising this point. We acknowledge that reducing resolution and using HOG features may lead to the loss of fine-grained details, which could impact tasks requiring precise texture or color information. However, our primary goal is to improve the efficiency and practicality of pretraining for vision classification tasks, where such fine details often contribute less to downstream performance compared to the overall structure and layout of the image. Our experiments demonstrate that, despite this potential information loss, the performance on tasks such as ImageNet classification remains on par with or better than previous methods.
>
> **Q2: Task-Specific Adaptability & Method Generality**
>
> A2: We appreciate the concern about task-specific adaptability. Our work focuses on a specific class of general vision tasks, particularly classification, segmentation, and detection, where HOG features perform robustly. We agree that tasks such as image generation or super-resolution reconstruction might demand finer details like texture and color information. While our current approach may not be directly applicable to such tasks, we believe it provides a solid foundation for efficient pretraining in its target domain. Given that task-specific adaptability is not the primary focus of our current research, we have incorporated a corresponding discussion in Line 542-552 based on your comment.

---

### Meta-Review · Area_Chair_6hWn · 2024-12-17

**Metareview:**

This submission received three negative sores and two positive scores after rebuttal. The major concern is about the novelty of the proposed approach. After carefully reading the paper, the review comments, the AC deemed that the paper should undergo a major revision and promotion, thus is not ready for publication in the current form.

**Additional Comments On Reviewer Discussion:**

After discussion, reviewer #cXUf thought that `the conclusions drawn from the empirical study in this paper are not particularly innovative`. The reviewer# kS5H and teGB had no response to the rebuttal while keeping the negative scores.

---

### Decision · Program_Chairs · 2025-01-22

Reject